# Syncytin-mediated open-ended membrane tubular connections facilitate the intercellular transfer of cargos including Cas9 protein

Congyan Zhang, Randy Schekman*

Department of Molecular and Cell Biology, Howard Hughes Medical Institute, University of California, Berkeley, United States

**Abstract** Much attention has been focused on the possibility that cytoplasmic proteins and RNA may be conveyed between cells in extracellular vesicles (EVs) and tunneling nanotube (TNT) structures. Here, we set up two quantitative delivery reporters to study cargo transfer between cells. We found that EVs are internalized by reporter cells but do not efficiently deliver functional Cas9 protein to the nucleus. In contrast, donor and acceptor cells co-cultured to permit cell contact resulted in a highly effective transfer. Among our tested donor and acceptor cell pairs, HEK293T and MDA-MB-231 recorded optimal intercellular transfer. Depolymerization of F-actin greatly decreased Cas9 transfer, whereas inhibitors of endocytosis or knockdown of genes implicated in this process had little effect on transfer. Imaging results suggest that intercellular transfer of cargos occurred through open-ended membrane tubular connections. In contrast, cultures consisting only of HEK293T cells form close-ended tubular connections ineffective in cargo transfer. Depletion of human endogenous fusogens, syncytins, especially syncytin-2 in MDA-MB-231 cells, significantly reduced Cas9 transfer. Full-length mouse syncytin, but not truncated mutants, rescued the effect of depletion of human syncytins on Cas9 transfer. Mouse syncytin overexpression in HEK293T cells partially facilitated Cas9 transfer among HEK293T cells. These findings suggest that syncytin may serve as the fusogen responsible for the formation of an open-ended connection between cells.

*For correspondence:
schekman@berkeley.edu

## Editor's evaluation

This fundamental work extends and in substantive ways introduces new concepts in the mode of communication between cells. Molecules pass from one cell to another through membrane tubules and the investigators show here convincingly that this occurs exclusively through the physical connection of the open ended tubules and not through exosomes; this process requires syncytin proteins, and the functionality of the protein transferred is retained.

## Introduction

Although most forms of intercellular communication are mediated by secreted diffusible proteins and small molecules, considerable interest has developed concerning the possibility that extracellular vesicles (EVs) or exosomes and tubular connections may convey cytoplasmic proteins, RNA molecules, and even organelles between neighboring or distant cells (*Aykan, 2013*).

Exosomes, one type of EV, are secreted from diverse cells and have been reported to elicit physiologically meaningful effects on target cells (*van Niel et al., 2018*; *Mathieu et al., 2019*). Exosomes contain various biological molecules, including lipids, RNA, and proteins, any or all of which may

**eLife digest** Communication between cells is an important process for survival, especially in multicellular organisms. Cells typically exchange information by releasing small molecules in to their surrounding environment which neighboring cells then receive and respond to. However, there is growing evidence to suggest that cells also pass signals to each other via fatty bubbles called exosomes and tubes connecting their membranes.

Various reports have suggested that these mechanisms can transport larger proteins and nucleic acids which carry the information cells need to make proteins. However, how cells are able to combine their membranes to allow these types of transfer is unclear.

To investigate, Zhang and Schekman studied how human cancer cells and embryonic cells grown in a laboratory pass molecules between each other. This included a string of nucleic acids known as RNA and a protein called Cas9 which can edit the genome of cells to activate an enzyme that has bioluminescence activity. By measuring the level of luminescence, Zhang and Schekman were able to sensitively detect the transfer of Cas9 and RNA to neighboring cells.

The experiments showed that exosomes were not efficient at transporting proteins or RNA. However, cells in near or direct contact transferred both molecules effectively using tube connections, with some cell types being more adept at this mechanism than others. Zhang and Schekman found that the formation of these tubular channels required a protein called syncytin which helps membranes fuse together mainly in the early stages of embryo development.

These findings open a new avenue of investigation on how cells send signals to one another. It is also possible that the protein syncytin has a role in cancer progression, as tumors rely on cell communication to maintain their growth and organize the cells surrounding them. However, further work is needed to investigate this possibility.

introduce normal or pathological signals to a target cell or tissue (*van Niel et al., 2018*; *Mathieu et al., 2019*). Our previous studies reported that two RNA-binding proteins, YBX1 and Lupus La protein, are involved in highly selective microRNA loading into exosomes (*Shurtleff et al., 2016*; *Temoche-Diaz et al., 2019*). The use of exosomes as a diagnostic biomarker and as a delivery vehicle for drugs or the CRISPR/Cas9 has generated much interest (*Kim et al., 2017*; *Li et al., 2019*; *Lin et al., 2018*; *de Jong et al., 2020*). We know little about the means by which exosomes may efficiently deliver cargo to the cytoplasm or nucleus of cells that internalize such vesicles (*Kalluri and LeBleu, 2020*). Functional delivery will likely require the action of a membrane fusogen to promote fusion of an exosome at the cell surface or after internalization to an endosome.

The tunneling nanotube (TNT) is a membranous structure with the potential for intercellular communication (*Rustom et al., 2004*). TNTs are thin membrane bridges with open-ended extremities that appear to mediate membrane continuity between cultured mammalian cells (*Rustom et al., 2004*). TNTs were first described in cultured rat pheochromocytoma PC12 cells, and although considerable evidence has developed for intercellular transfer mediated by TNTs in cell culture, the possibility of a physiological role needs to be further explored (*Pinto et al., 2020*). Diverse cargo, including small organelles of the endosomal/lysosomal system and mitochondria, calcium, MHC class I proteins, miRNAs, mRNAs, prions, viral and bacterial pathogens, have been found to traverse open-ended TNT connections among a variety of cells (*Haimovich et al., 2017*; *Eugenin et al., 2009*; *Gerdes, 2009*; *Gerdes et al., 2007*; *Gerdes and Carvalho, 2008*; *Gurke et al., 2008*; *Wang et al., 2010*; *Kimura et al., 2012*; *Kolba et al., 2019*). TNTs are usually 50–1000 nm in width; however, thicker tubules of 1–2 μm in width, called tumor microtubes, have been reported in cultures of cancer cells (*Roehlecke and Schmidt, 2020*). Tumor microtubes may play roles in the tumor microenvironment, especially glioblastoma (*Osswald et al., 2015*; *Venkataramani et al., 2022*). This emerging form of intercellular tubular connections may enhance our understanding of cellular community in multicellular organisms and in disease progression (*Yamashita et al., 2018*). As with exosomes, open-ended tubular connections between cells must involve a membrane fusogen to promote intercellular traffic of proteins, RNA, and organelles.

Cell-cell fusion serves vital roles in virtually all organisms from yeast to mammals and plants (*Chen and Olson, 2005*), Mammals have a limited number of known catalysts of cell fusion that serve

specialized cells and tissues. Two examples are myomaker and myomerger for myoblast fusion, and the syncytins for trophoblast cell fusion (reviewed in *Brukman et al., 2019*). Syncytins are endogenous retroviral envelope glycoproteins: syncytin-1 and syncytin-2 in humans (*Blaise et al., 2003*; *Frendo et al., 2003*; *Mi et al., 2000*) and syncytin-A and -B in mice (*Dupressoir et al., 2005*; *Peng et al., 2007*). Syncytin-mediated fusion requires complementary cell surface receptors, including ASCT-2, a widely distributed neutral amino acid transporter, a receptor for syncytin-1 (*Blond et al., 2000*) and Major Facilitator Superfamily Domain Containing-2 (MFSD2), a sodium-dependent lysophospha-tidylcholine transporter, the receptor for syncytin-2 (*Esnault et al., 2008*). In the mouse, lymphocyte antigen 6E (Ly6e) has been identified as a receptor for syncytin-A (*Bacquin et al., 2017*). In addition to a role in the formation of a trophoblast syncytium during early embryonic development, syncytins are also involved in the fusion of osteoclast precursors and cancer cells (*Bjerregaard et al., 2006*; *Uygur et al., 2019a*).

Cytoplasmic structure-forming proteins are required to organize the plasma membrane as cells adhere in preparation for fusion. For example, invasive protrusions that promote the cell membrane juxtaposition and fusogen engagement are propelled by Arp2/3-mediated branched actin polymerization (*Sens et al., 2010*; *Shilagardi et al., 2013*). In this process, dynamin, a large GTPase best known for its role in endocytosis, bundles actin filaments to promote invasive protrusion formation (*Zhang et al., 2020*).

Here, we report the use of quantitative assays to measure traffic of protein and RNA cargo between cells. A Cas9/gRNA cargo enriched in exosomes was found to be inefficiently delivered for gene editing in a reporter cells. Similar results have been reported elsewhere (*de Jong et al., 2020*; *Haimovich et al., 2017*; *Somiya and Kuroda, 2021*; *Albanese et al., 2021*). In contrast, we found highly efficient transfer when cells were co-cultured with physical contact. Using live-cell imaging and correlative light and electron microscopy (CLEM), we found that traffic appeared to be mediated by open-ended membrane tubular connections of several microns in diameter. Membrane fusion depended on syncytin in the acceptor cell and a complementary receptor in the donor cell. Optimum transfer was seen with HEK293T cells as a donor and MDA-MB-231 cells and several other tumor cell lines as acceptor. In contrast, HEK293T cells serving as both a donor and an acceptor were not active in intercellular traffic but instead appeared to form close-ended tubular connections.

## Results

### Exosome-mediated Cas9 intercellular transfer is inefficient

To determine whether exosomes efficiently deliver cargo proteins and RNA to recipient cells, we developed a Cas9-based dual-luciferase cargo reporter assay. In donor cells, a modified 'retention using a selective hook (RUSH)' strategy (*Boncompain et al., 2012*) was applied to Cas9 tethered indirectly to CD63, a tetraspanin membrane protein enriched in endosomes and exosomes. CD63 was fused with streptavidin, and Cas9, to which a nuclear localization signal was appended, was fused to streptavidin-binding protein (SBP). gRNA expression was driven separately by a U6 promoter (*Figure 1A*). These two chimeric proteins associate noncovalently and dissociate in the presence of biotin (*Figure 1—figure supplement 1A*). HEK293T cells were stably transfected with one or both fusion genes and secreted exosomes were purified by differential and buoyant density centrifugation. Exosomes from cells expressing both fusion proteins were enriched in Cas9 and gRNA, whereas cells expressing Cas9 untethered to CD63 had little or no Cas9 or gRNA (*Figure 1B and C*). Protease and nuclease protection experiments showed that Cas9 and the gRNA were both sequestered within vesicles (*Figure 1—figure supplement 1B and C*).

In creating a quantitative and sensitive Cas9 reporter cell line, we tried variations on a method described elsewhere (*de Jong et al., 2020*; *Gee et al., 2020*) and settled on an approach involving editing of an out-of-frame copy of the nanoluciferase gene (Nluc) in different target cells, HEK293T, U2OS, or MDA-MB231 cell lines (*Figure 1D*, *Figure 1—figure supplement 2*). In this approach, constitutively expressed firefly luciferase (Fluc) was expressed linked to the sequence of a self-cleaving peptide, F2A, followed by a Cas9/gRNA-targeted linker region and a stop codon. The Nluc sequence was placed after the stop codon with one nucleotide out of frame such that Nluc expression would depend upon cleavage and error-prone non-homologous end joining to remove the stop codon and restore a proper reading frame. Synthesis of the gRNA was driven independently by a U6 promoter.

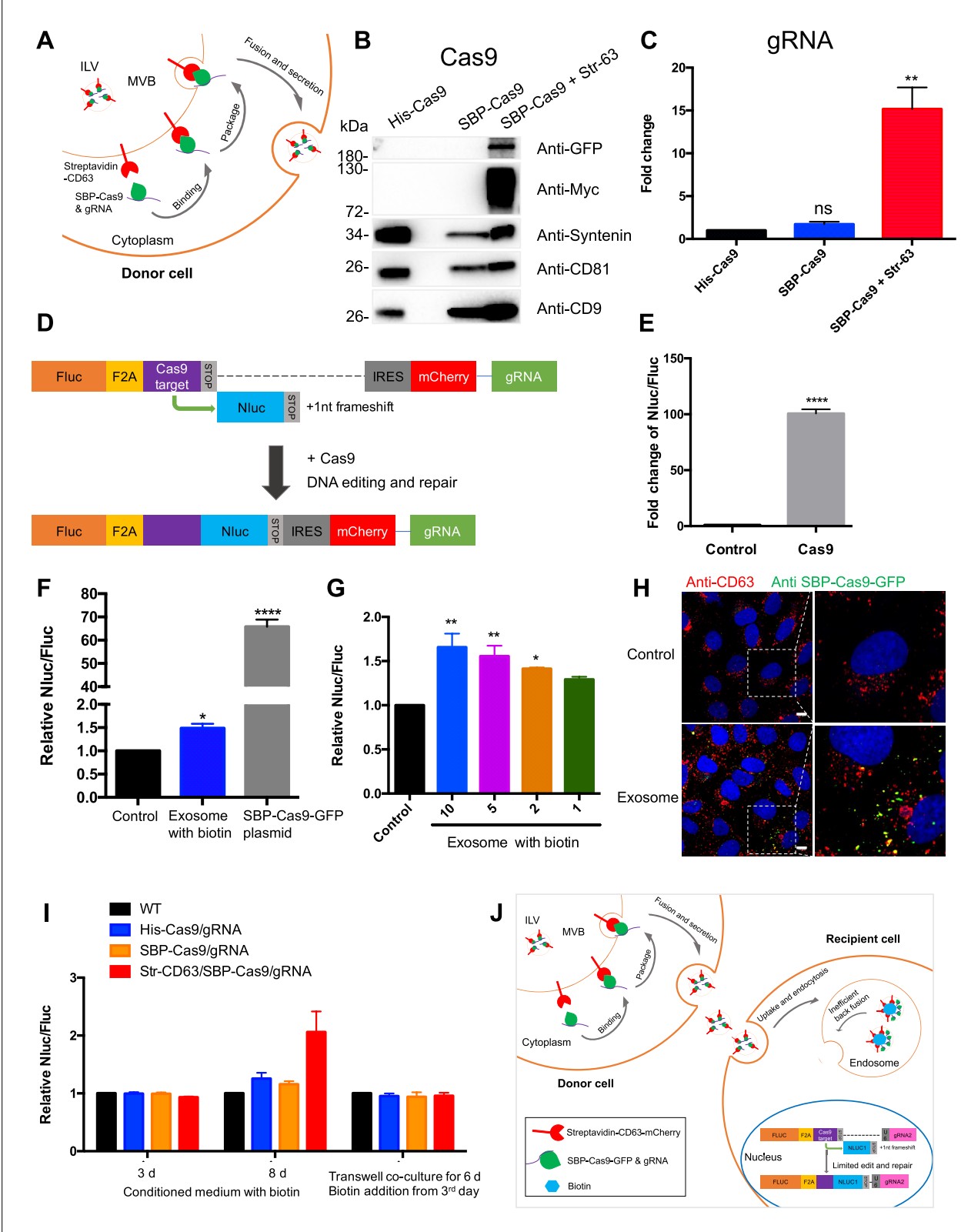

**Figure 1.** Exosome-mediated Cas9 intercellular transfer is inefficient. (**A**) Schematic showing how the modified RUSH system was used for packaging Cas9/gRNA into exosomes. (**B**) His-tagged and Flag-tagged Cas9-GFP fusion protein were expressed in HEK293T cells stably as a negative control. Exosomes from the three stable cell lines (His-Flag-Cas9-GFP, SBP-Flag-Cas9-GFP only, or SBP-Flag-Cas9-GFP and Myc-streptavidin-CD63-mCherry) were purified. Cas9-GFP protein was detected in exosomes from the cells expressing both SBP-Flag-Cas9-GFP and Myc-streptavidin-CD63-mCherry.

*Figure 1 continued on next page*

*Figure 1 continued*

(**C**) gRNA in exosomes from the three stable cell lines was quantified. gRNA in exosomes from cells with SBP-Flag-Cas9-GFP and Myc-streptavidin-CD63-mCherry was enriched ~15× with respect to cells His-Flag-Cas9-GFP only. Data represent mean ± SEM, n ≥ 3. ns, not significant, **p<0.01, one-way ANOVA. (**D**) Schematic representation of the reporter system. Firefly luciferase gene (Fluc) expressed constitutively followed by a Cas9/gRNA-targeted linker region and a stop codon. The Nanoluc gene (Nluc) was placed after the stop codon with one nucleotide out of frame; thus, Nluc cannot be expressed without Cas9 editing. After Cas9/gRNA is expressed or transferred, the Cas9/gRNA target linker region may be cleaved and subsequent DNA repair via non-homologous end joining may induce a frameshift in the linker region to restore some Nluc gene expression. (**E**) Proof of concept. In HEK293T cells with the reporter plasmid, the expression of Cas9 protein increased the Nluc/Fluc signal dramatically compared to that transfected with control empty plasmid. Data represent mean ± SEM, n ≥ 3. ****p<0.0001, two-tailed *t*-test. (**F**) The engineered exosomes were incubated with the reporter cells for 24–48 hr, and the cells were washed for the detection of Nluc/Fluc. SBP-Cas9-GFP plasmid was introduced by transfection as a positive control. Data represent mean ± SEM, n ≥ 3. *p<0.05, ****p<0.0001, one-way ANOVA. (**G**) Different amounts of engineered exosomes were incubated with reporter cells for detection of the Nluc/Fluc signal. Data represent mean ± SEM, n ≥ 3. *p<0.05, **p<0.01, one-way ANOVA. (**H**) Engineered exosomes from HEK293T cells were incubated with U2OS cells for 16 hr followed by immunofluorescence detection using anti-GFP (green) and anti-CD63 (red) antibodies. The nucleus was stained by Hoechst 33342. Scale bar is 10 μm. (**I**) The conditioned medium from different donor cells was used to culture the reporter cells in the presence of 40 μM biotin, or the donor cells and the recipient cells were co-cultured for 6 days in a transwell dish (0.45 μm pore). Biotin was added from the third day followed by Nluc/Fluc assay after 6 days. (**J**) A proposed model suggesting that exosome-mediated Cas9 intercellular transfer is inefficient.

The online version of this article includes the following source data and figure supplement(s) for figure 1:

**Source data 1.** Uncropped Western blot images corresponding to *Figure 1B*.

**Figure supplement 1.** Validation of modified RUSH strategy.

**Figure supplement 1—source data 1.** Uncropped Western blot images corresponding to *Figure 1—figure supplement 1A*.

**Figure supplement 1—source data 2.** Uncropped Western blot images corresponding to *Figure 1—figure supplement 1B*.

**Figure supplement 2.** Cas9-based reporter optimization.

Expression of the upstream Fluc gene served as the denominator in a measure of the efficiency of Nluc repair. As a test, we expressed Cas9 protein in HEK293T cells with the reporter plasmid and found that expression of Cas9 increased the Nluc/Fluc signal around 100-fold compared to that transfected with control empty plasmid (*Figure 1E*).

Exosomes secreted from the donor cell line were purified as described earlier and incubated with the reporter cell line in the presence of exogenous biotin to dissociate Cas9 from the CD63 tether. At an excess of exosomes/cell of $10^5/1$, the Nluc/Fluc ratio increased by only 50%, markedly lower than the 70-fold increase in the ratio of reporter cells directly transfected with the Cas9/gRNA construct (*Figure 1F*) or co-transfected with both SBP-Cas9/gRNA and streptavidin-CD63 in the presence or absence of biotin (*Figure 1—figure supplement 1D*). Nonetheless, the signal was somewhat proportional to dose of exosomes (*Figure 1G*), but we concluded that functional delivery of Cas9/gRNA from exosomes secreted by HEK293T cells was very inefficient. Similar experiments were performed using Cas9-enriched exosomes secreted by MDA-MB231 cells, and we found that these exosomes were also inefficient for the Cas9/gRNA delivery to MDA-MB231 cells as a reporter (*Figure 1—figure supplement 1E and F*). Our findings were similar to those reported previously (*de Jong et al., 2020*; *Haimovich et al., 2017*; *Somiya and Kuroda, 2021*).

In a control experiment, we used exosomes enriched in Cas9-GFP secreted by HEK293T cells to monitor internalization into U2OS cells. Tagged GFP antibody was used to detect internalized Cas9-GFP that accumulated in intracellular puncta in ~30% of the cells during a 16 hr incubation (*Figure 1H*). From these experiments, we concluded that Cas9 was internalized into cells but may not have been mobilized from vesicles to gain access to the luciferase reporter genes in the nucleus.

We considered the possibilities that the secreted exosomes were fragile for storage or damaged during the purification procedure. Unfractionated conditioned medium from the donor cells was incubated with reporter cells for 3–8 days with little change in the Nluc/Fluc signal (*Figure 1I*). As a further test of vesicle stability, we incubated donor and acceptor cells in a transwell chamber separated by a 0.45 μm pore vesicle-permeable membrane. After a 6-day co-culture, no increase in the Nluc/Fluc was observed (*Figure 1I*). Therefore, we suggest delivery of Cas9 is limited by inefficient fusion of exosomes with the endosomal membrane (*Figure 1J*).

## Intercellular transfer of Cas9 and other cargos through direct cell-cell contact

We next tested the possible transfer of Cas9/gRNA mediated by cell-cell contact. In the initial experiments, we used donor HEK293T cell lines expressing His-tagged Cas9-GFP/gRNA (His-Cas9/gRNA), SBP-tagged Cas9-GFP/gRNA (SBP-Cas9/gRNA), and SBP-Cas9-GFP/gRNA+Myc-streptavidin-CD63-mCherry (Str-CD63/SBP-Cas9/gRNA). As acceptor cell lines with an integrated reporter cassette, we used HEK293T, U2OS, MDA-MB-231, A549, and MCF7. After a co-culture for 6 days to near confluence, we observed substantial Nluc expression in all combinations except where HEK293T cells served as both donor and acceptor (*Figure 2A*). The highest level of Nluc expression (~60-fold increase in the Nluc/Fluc signal) was seen in co-culture of HEK293T and MDA-MB-231 cells. Of note, we found little difference in the efficiency of transfer with CD63-tethered or free Cas9/gRNA. In a time-course experiment, we observed progressive transfer over several days with a limit reached after day 3 (*Figure 2—figure supplement 1A*). On increase in the donor:acceptor cell ratio, we observed a gradual increase in the signal at a higher level of donor cell but HEK293T cells serving as both donor and acceptor remained inactive even at a donor:acceptor ratio of 10:1 (*Figure 2—figure supplement 1B and C*).

As an independent test of intercellular transfer, we established a quantitative approach using split-GFP and two other markers of donor and acceptor cells (*Figure 2B*). In this approach, we expressed CFP and CD63 fused to 7*tandem GFP11 in cell-1, and mCherry and GFP1-10 in cell-2. A mixture of cells before co-culture did not exhibit a GFP signal (*Figure 2—figure supplement 1D and E*). On co-culture and transfer to form GFP, the cell types would be distinguished by predominant CFP and mCherry signals in cells 1 and 2, respectively (*Figure 2B*). Flow cytometry analysis of the HEK293T and MDA-MB-231 cells separately showed distinct signals of CFP and mCherry, respectively (*Figure 2C*, top panel Q4-4 quadrant; middle panel, P3 area). After co-culture, doubl- positive cells were found (*Figure 2C*, bottom panel, Q2 and Q2-4 quadrants), suggesting transfer of one or both fragments of GFP between cells (*Figure 2C*). GFP+mCherry+/mCherry+ (*Figure 2C*, bottom panel) represented the ratio of cells with GFP+mCherry double-positive fluorescence, indicating a transfer of GFP11 from HEK293T to MDA-MB-231 in around 6% of cells (*Figure 2D and E*, yellow bars). GFP+CFP+/CFP+ (*Figure 2C*, bottom panel) represented the ratio of cells with GFP+CFP double-positive fluorescence indicating transfer from MDA-MB-231 to HEK293T in around 0.5% of cells (*Figure 2D and E*, green bars). As a control, HEK293T cells expressing CFP and CD63 fused 7*tandem GFP11 (*Figure 2C*, top panel) were co-cultured with HEK293T cells expressing mCherry and GFP1-10 (*Figure 2F*, top panel, P3 area). No cargo transfer was seen in this combination (*Figure 2F*, bottom panel, Q2 and Q2-4 quadrants). One complication was the appearance of a double-positive signal of CFP and mCherry that we attributed to cell adhesion (*Figure 2F*, bottom panel, Q2-5 quadrant with asterisk).

Co-cultures of MDA-MB-231 as the donor to several other tumor cell lines as acceptor (ratio of 10:1) resulted in cargo transfer to a limit of about 10% of cells (*Figure 2G*). Of note, cargos were also transferred from MDA-MB-231 to other cell lines, including the MDA-MB-231 cell line itself (GFP+CFP+/CFP+ bars in *Figure 2G*). In summary, by two independent measures, we conclude that intercellular transfer of proteins occurs in certain donor–acceptor cell pairs dependent on cell-cell contact.

We used the split GFP assay as an independent means to assess the efficiency of transfer of a cytoplasmic protein sequestered in exosomes to the cytoplasm of an acceptor cell. Exosomes were isolated from donor cells expressing CD63 fused to the 7*tandem GFP11 and incubated at a ratio of $10^5/1$ with acceptor cells expressing GFP10 for 12–24 hr. In contrast to the ~6% of cells that received GFP11 from donor cells in a 3–4-day co-culture, only 0.02% of acceptor cells reported GFP in the 12–24 hr incubation with exosomes (*Figure 2D*). These results reinforce the suggestion that the delivery of exosome content may be limited by inefficient fusion to the endosomal membrane at least for exosomes produced by HEK293T and internalized by MDA-MB-231 cells.

## F-actin but not endocytosis required for intercellular transfer of Cas9

We considered the possibility that cell contact may be required to organize the directed transfer of extracellular vesicles between donor and acceptor cells. If so, vesicles derived at a junction between donor and acceptor cells may be endocytosed locally. As a simple test of the possibility, we compared the effect of a variety of endocytosis inhibitors on transfer of Cas9/gRNA between cells. Chlorpromazine, LY294002, and wortmannin were found not to interfere with Cas9 transfer at concentrations where the internalization of a fluorescently tagged transferrin or zymosan was blocked (*Figure 3A*,

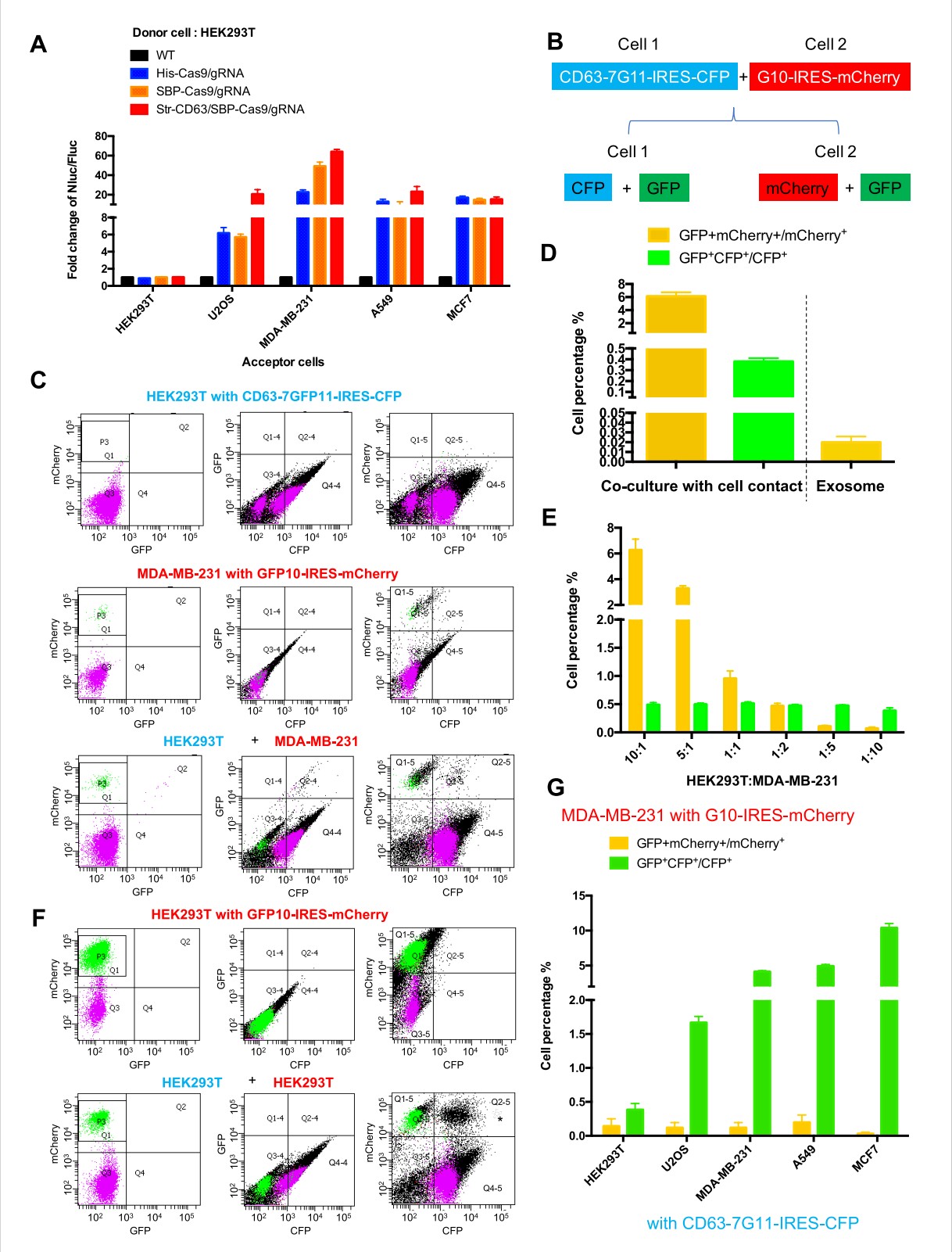

**Figure 2.** Intercellular transfer of Cas9 and other cargos through direct cell-cell contact. (**A**) Donor cells: HEK293T wild-type (WT) with stable overexpression of his tagged Cas9-GFP/gRNA (His-Cas9/gRNA), with stable overexpression of SBP tagged Cas9-GFP/gRNA (SBP-Cas9/gRNA) or with stable overexpression of SBP-Cas9-GFP/gRNA and Myc-streptavidin-CD63-mCherry (Str-CD63/SBP-Cas9/gRNA). The last construct permits Cas9 incorporation into endosomes as depicted in *Figure 1J*. Acceptor cells: HEK293T, U2OS, MDA-MB-231, A549, or MCF7 with stable transfection of

*Figure 2 continued on next page*

*Figure 2 continued*

the reporter plasmid. After 6 days of co-culture, Nluc/Fluc assays were performed and normalized to an aliquot of co-cultured WT donor and reporter cells. (**B**) Diagram showing trifluorescence split-GFP system for the detection of intercellular transfer. (**C**) HEK293T expressing CD63 fused 7-tandem GFP11 and CFP (top) was co-cultured with MDA-MB-231 expressing GFP1-10 and mCherry (middle), after 3 days the co-cultures were analyzed by flow cytometry (bottom). The plots are displayed in an all-cell mode. The possible singlet and doublet are indicated with green/purple dots and black dots, respectively. Quadrants Q2, Q2-4, Q2-5 mainly represent double-positive. (**D**) Double-positive fluorescent cells were quantified. GFP$^+$ mCherry$^+$/ mCherry$^+$ represents the ratio of cells with GFP+mCherry double-positive fluorescence to cells with mCherry; GFP$^+$CFP$^+$/CFP$^+$ represents the ratio of cells with GFP+CFP double-positive fluorescence to cells with CFP. Three independent experiments were performed. Data represent mean ± SEM. (**E**) HEK293T expressing CD63 fused 7*tandem GFP11 and CFP was co-cultured with MDA-MB-231 expressing GFP1-10 and mCherry. The ratio of HEK293T to MDA-MB-231 is indicated. After 3 days, the co-cultures were analyzed by flow cytometry and double-positive fluorescence was quantified. Data in this figure represent mean ± SEM, n ≥ 3. (**F**) HEK293T expressing CD63 fused 7-tandem GFP11 and CFP (same as **C**, top) was co-cultured with HEK293T expressing GFP1-10 and mCherry (top) and after 3 days the co-cultures were analyzed by flow cytometry (bottom). The plots are displayed in an all-cell mode. The possible singlet and doublet are indicated with green/purple dots and black dots, respectively. Quadrants Q2, Q2-4, Q2-5 mainly represent double-positive. Asterisk in the bottom panel indicates that the double positive of mCherry and CFP may derive from adherent cells, not intercellular transfer. (**G**) MDA-MB-231 expressing GFP1-10 and mCherry was co-cultured with other cell lines expressing CD63 fused 7-tandem GFP11 and CFP at the ratio of 10:1 and after 3 days the co-cultures were analyzed by flow cytometry and double-positive fluorescence was quantified. Three independent experiments were performed. Data represent mean ± SEM.

The online version of this article includes the following figure supplement(s) for figure 2:

**Figure supplement 1.** The effects of co-culture time and different ratios of donor to recipient cells on transfer.

*Figure 3—figure supplement 1A–D*). Furthermore, siRNA knockdown of clathrin heavy chain and AP-2 subunit B1 (validations in *Figure 3—figure supplement 2*) blocked endocytosis but had only a slight effect on Cas9 transfer compared to a control (*Figure 3B*) that dramatically blocked the uptake of transferrin by recipient cells (*Figure 3—figure supplement 1E and F*). For comparison, we also used siRNA to knockdown caveolin and flotillin 2 with little effect on endocytosis of transferrin but which stimulated transfer of Cas9 (*Figure 3B*, *Figure 3—figure supplement 1E and F*). Endocytosis, at least as revealed by these particular drug sensitivities and coat protein requirements, appeared not to be necessary for Cas9/gRNA transfer.

As an alternative, we considered the possibility that intercellular transport was mediated by the formation of membrane tubular connections. These structures, which have been reported in many cultured cell lines, appear to be formed and stabilized by transcellular actin filaments (*Yamashita et al., 2018*). We tested the effect of several latrunculins (Lat A/B) that depolymerize F-actin and of actin and Arp2/3 shRNA knockdowns. Lat A (80 or 200 nM) and LatB (2.5 or 5 µM) substantially blocked the transfer of Cas9 (*Figure 3C*). Actin shRNA knockdown in donor HEK293T cells reduced Cas9 transfer threefold but was more effective in blocking transfer in acceptor MDA-MB-231 cells (*Figure 3D and E*), which may be because the actin knockdown efficiency was greater in MDA-MB-231 than in HEK293T cells (*Figure 3—figure supplement 2E and H*). Arp2/3 complex knockdown in donor but not in recipient cells reduced Cas9 transfer (*Figure 3D and E*). The formin inhibitor SMIFH2 decreased Cas9 transfer dramatically (*Figure 3F*). These results suggested the transfer of Cas9 protein from HEK293T to MDA-MB-231 depends upon actin and to some extent on the Arp2/3 complex or formin possibly to stabilize a cellular structure essential for intercellular traffic.

## Open-ended membrane tubular connections bridge cell-cell communication

We sought a means to visualize intercellular membrane tubular connections using differentially tagged donor and acceptor cells. For live-cell imaging, HEK293T cells containing SBP-Cas9-GFP and Myc-streptavidin-CD63-mCherry were co-cultured with MDA-MB-231 wild-type cells. Mixed cultures were incubated with fluorescent wheat germ agglutinin (WGA), which labeled both cells and was used principally to mark the cell surface and endolysosome network of MDA-MB-231 cells. Membrane tubes appearing as pseudopodia or filopodia projected from HEK293T cells were seen to make contact with MDA-MB-231 cells forming what appeared to be open-ended connections (*Figure 4A*). On prolonged inspection, puncta containing endosome-related vesicles (labeled with streptavidin-CD63-mCherry) and SBP-Cas9-GFP were transported from HEK293T to MDA-MB-231 cells (*Figure 4B*), consistent with open-ended tubular connections associated with intercellular traffic. Such structures were seen to form and break during the course of visualization

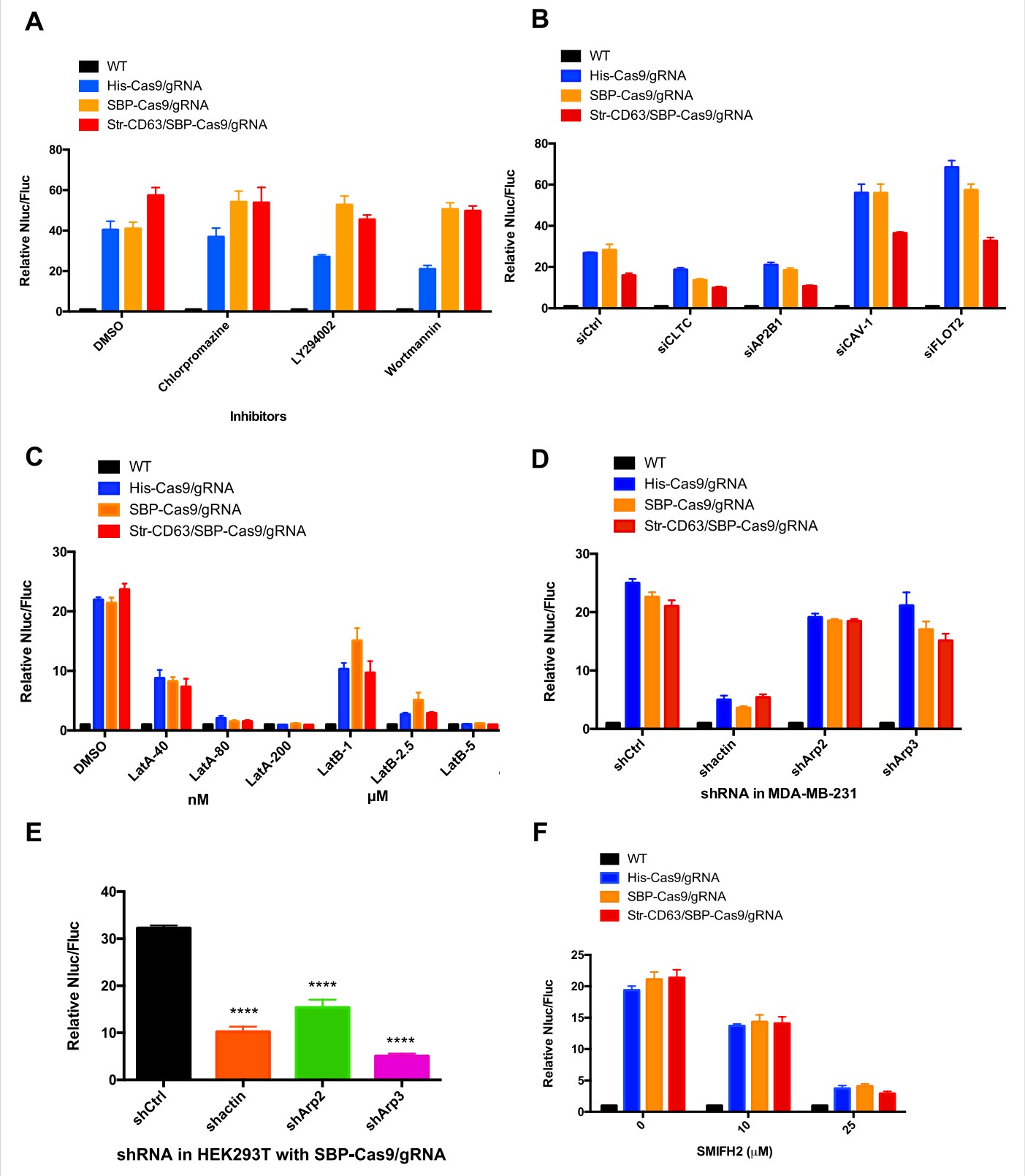

**Figure 3.** Inhibitors of F-actin but not of endocytosis reduce intercellular transfer of Cas9. (**A–D**, **F**) Donor cells: HEK293T wild-type (WT) with stable overexpression of his tagged Cas9-GFP/gRNA (His-Cas9/gRNA), with stable overexpression of SBP tagged Cas9-GFP/gRNA (SBP-Cas9/gRNA) or with stable overexpression of SBP-Cas9-GFP/gRNA and Myc-streptavidin-CD63-mCherry (Str-CD63/SBP-Cas9/gRNA). The recipient cell line was MDA-MB-231 with a reporter plasmid. Nluc/Fluc assays were performed and normalized to an aliquot of co-cultured WT donor and reporter cells. (**A**) Nluc/

*Figure 3 continued on next page*

*Figure 3 continued*

Fluc activities were measured after donor cells and acceptor cells were co-cultured for 3 days with DMSO or different inhibitors. (**B**) The indicated genes were knocked down in recipient cells via siRNA and then co-cultured with donor cells for 3 days. siCtrl represents a negative control for the siRNA knockdown. CLTC, clathrin heavy chain; AP2B1, adaptor-related protein complex 2 subunit beta 1; CAV-1, caveolin 1; FLOT2, flotillin 2. (**C**) The donor cells and acceptor cells were co-cultured for 3 days with either DMSO, 40, 80, or 200 nM latrunculin A (LatA), or 1, 2.5, or 5 μM latrunculin B (LatB). The Nluc/Fluc signal detected after co-culture suggested that more than half of the cells remained viable during drug treatment. (**D**) The indicated genes were knocked down in recipient cells via shRNA that were then co-cultured with donor cells for 3 days. siCtrl represents negative control for the siRNA knockdown. (**E**) Donor cells: HEK293T with stable overexpression of SBP tagged Cas9-GFP/gRNA (SBP-Cas9/gRNA). The recipient cell line was MDA-MB-231 with the reporter plasmid. The indicated genes were knocked down in donor cells via shRNA that were then co-cultured with recipient cells for 3 days. (**F**) Donor cells and acceptor cells were co-cultured for 4 days with either DMSO, 10, or 25 μM formin inhibitor, SMIFH2. The Nluc/Fluc signal detected after co-culture suggested that more than 70% of the cells remained viable during drug treatment. Nluc/Fluc assays were performed and normalized to an aliquot of co-cultured WT donor and reporter cells. Data in this figure represent mean ± SEM, n ≥ 3. ****p<0.0001, one-way ANOVA.

The online version of this article includes the following source data and figure supplement(s) for figure 3:

**Figure supplement 1.** Endocytosis inhibitors and endocytosis protein knockdown block transferrin uptake.

**Figure supplement 2.** Knockdown validation.

**Figure supplement 2—source data 1.** Uncropped Western blot images corresponding to *Figure 3—figure supplement 2A*.

**Figure supplement 2—source data 2.** Uncropped Western blot images corresponding to *Figure 3—figure supplement 2B*.

**Figure supplement 2—source data 3.** Uncropped Western blot images corresponding to *Figure 3—figure supplement 2C*.

**Figure supplement 2—source data 4.** Uncropped Western blot images corresponding to *Figure 3—figure supplement 2D*.

**Figure supplement 2—source data 5.** Uncropped Western blot images corresponding to *Figure 3—figure supplement 2E and H*.

**Figure supplement 2—source data 6.** Uncropped Western blot images corresponding to *Figure 3—figure supplement 2F and I*.

**Figure supplement 2—source data 7.** Uncropped Western blot images corresponding to *Figure 3—figure supplement 2G and J*.

(*Figure 4—video 1*); however, in spite of a membrane fusion event, we did not observe whole-sale cell fusion to form heterokaryons. At the same time, many much thinner projections, termed tunneling nanotubes, formed between neighboring cells. The open-ended tubular connections were also observed using z-stack imaging and 3D tomography (*Figure 5A and B*, *Figure 5—videos 1 and 2*), confirming the observation of open-ended tubular connections between HEK293T and MDA-MB-231 cells.

We used CLEM to provide a closer inspection of tubular connections between donor and acceptor cells. Candidate cell pairs observed by confocal microscopy (*Figure 5C*) were fixed, stained, and processed for thin section EM (*Figure 5D*). Serial images were stacked to reveal entire tubular connections between HEK293T and MDA-MB-231 cells (one example in *Figure 5E*). A trace of the plasma membranes in connecting cells revealed a potential point of fusion and diverse organelles populating the junction between cells. The cell junctions that appeared to convey cargo in these cell pairs were much thicker, from 2 to 4 microns in diameter, than the 100–200 nanometer TNTs also seen in our mixed cultures. Combining the live-cell imaging and CLEM results, we found 8 open-ended tubular connections in 120 tubule-cell contacts, similar to the ratio of recipient cells containing GFP signal after co-culture in the split-GFP assays (*Figure 2D and E*).

Although membrane tubules were seen to emanate from HEK293T cells (*Figure 4*, *Figure 4—video 1*), no intercellular movement of Cas9 or GFP reporter fragments were seen in cultures of HEK293T donor and acceptor pairs (*Figure 2*). We performed live-cell imaging to determine whether tubules formed by HEK293T cells make contact with other cells in HEK293T donor:acceptor cultures. Tubular connections were observed and in cases where the connections remained intact, no evidence of cytoplasmic continuity was observed (*Figure 6A*, *Figure 6—video 1*), consistent with our failure to detect Cas9 transfer between cells in a culture of HEK293T. Next, we evaluated HEK293 donor:acceptor cell pairs by CLEM. One such example revealed a close-ended tubular contact (*Figure 6B–D*), suggesting a failure in plasma membrane fusion at the tubular junction between HEK293T cells. Furthermore, combining live-cell imaging and CLEM results, we found no open-ended tubular connections in 120 tubule-cell contacts among HEK293T cells.

Altogether, the above results demonstrated that cargos were transferred through open-ended tubular connections that would depend upon a plasma membrane fusion process where a fusogen catalyst may be contributed by the donor or acceptor plasma membrane.

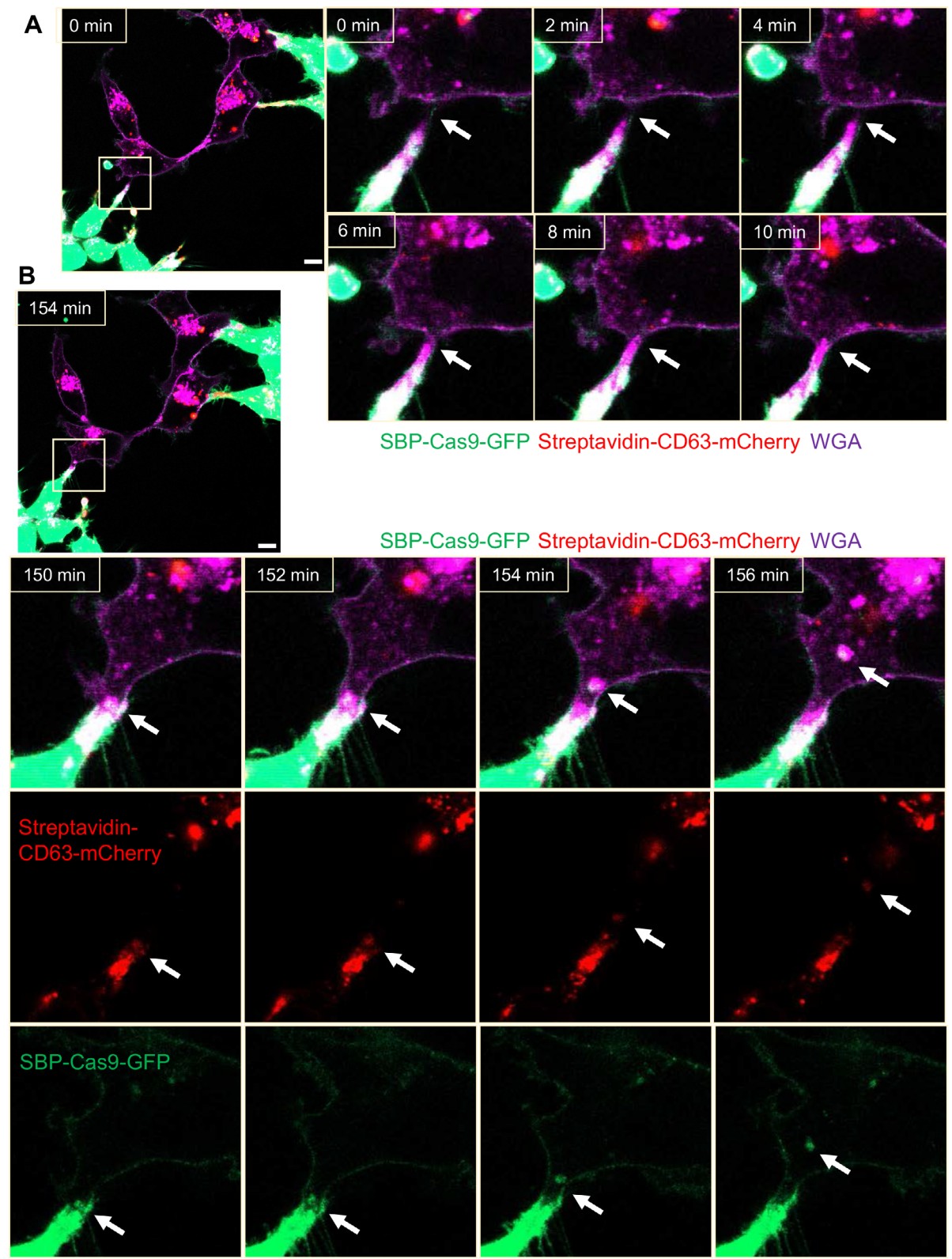

SBP-Cas9-GFP Streptavidin-CD63-mCherry WGA

**Figure 4.** Intercellular transfer of Cas9 protein through open-ended tubular connections. (**A, B**) Visualization of open-ended tubular connection structure and time-lapse imaging of the co-culture. Donor cells: HEK293T with stable overexpression of SBP-Cas9-GFP/gRNA and Myc-streptavidin-CD63-mCherry (Str-CD63/SBP-Cas9/gRNA). Acceptor cell: MDA-MB-231. Green: SBP-Cas9-GFP; red: Myc-streptavidin-CD63-mCherry; purple: CF640R WGA conjugate. Scale bar is 10 µm. The white arrows in (**A**) indicate the formation of an open-ended tubular connection. The white arrows in

*Figure 4 continued on next page*

*Figure 4 continued*

(**B**) indicate that the cargos including endosomal vesicles (red, middle row) and SBP-Cas9-GFP (green, bottom row) transferred through the open-ended tubular connection.

The online version of this article includes the following video and figure supplement(s) for figure 4:

**Figure supplement 1.** MDA-MB-231 and HEK293T wild-type cells were observed by confocal microscopy.

**Figure 4—video 1.** Intercellular transfer of Cas9 through open-ended tubular connections.

https://elifesciences.org/articles/84391/figures#fig4video1

## Syncytins in MDA-MB-231 mediate the formation of open-ended tubular connections

Humans have two virus-like fusogen proteins, syncytin-1 and syncytin-2, which are of retroviral origin and which serve a normal function in trophoblast cell fusion and possibly a pathological function in cancer cell fusion (*Frendo et al., 2003*; *Bjerregaard et al., 2006*). We examined the distribution and function of syncytin-1 and syncytin-2 in HEK293T cells and in the tumor cell lines in our Cas9 and GFP transfer assays. First, we used immunoblotting to detect syncytins in the cell lines used in this study (*Figure 7—figure supplement 1A*). The expression of syncytin-1 and syncytin-2 was detected in all these lines but the higher mobility, presumably furin processed form of syncytin-2, was more apparent than that of syncytin-1. MDA-MB-231 cells had a lower level of syncytin-1 (*Figure 7—figure supplement 1A*). CRISPRi was used to deplete syncytin-1 and syncytin-2 in MDA-MB-231 (*Figure 7—figure supplement 1B–E*). Because of the complication in using CRISPRi in a transfer assay that depended on the expression of Cas9, we titrated the expression of dCas9 and found that a low level did not itself affect the detection of functional Cas9 transfer from HEK293T to MDA-MB-231 cells (*Figure 7—figure supplement 1B–E*). Knockdown of syncytin-1 in MDA-MB-231 cells partially decreased Cas9 transfer; however, knockdown of syncytin-2 in MDA-MB-231 cells had a greater effect on Cas9 transfer (*Figure 7A*, *Figure 7—figure supplement 1B–E*, lane 2). Expression of the mouse syncytin, syncytin-A, partially restored Cas9 transfer in the syncytin-1 and syncytin-2 knockdowns (*Figure 7A*). The effect of syncytin-2 knockdown and the reversal by expression of mouse syncytin-A was particularly significant in the samples where His-tagged Cas9/gRNA was expressed in the donor HEK293T cells (*Figure 7A*).

In order to assess the functional importance of various N- and C-terminal domains of syncytin-A (*Peng et al., 2007*), we constructed truncated versions for expression in syncytin-2 knockdown MDA-MB-231 cells (*Figure 7B*, *Figure 7—figure supplement 1F*). As before, functional restoration of Cas9 transfer was observed by expression of full-length syncytin-A, but no significant Cas9 transfer was observed on expression of any of the syncytin-A truncated versions (*Figure 7C*). Indeed, expression of truncated syncytin-As reduced functional Cas9 transfer below the level achieved by knockdown of syncytin-2 alone (*Figure 7C*).

Next, we evaluated the effect of syncytin knockdown in HEK293T cells and found that the depletion of syncytin-1 or 2 in HEK293T reduced functional Cas9 transfer by ~30%, a much smaller effect than seen in depletion of syncytin-2 in MDA-MB-231 cells (*Figure 7D*, *Figure 7—figure supplement 1G and H*). The expression of syncytin-A in HEK293T cells serving as a reporter permitted some transfer of Cas9 from HEK293T cells serving as a donor (*Figure 7E*). Although the effect was much less pronounced than when MDA-MB-231 served as acceptor cells (up to 4-fold vs. 30-fold change of Nluc/Fluc signal indicating Cas9 transfer), this result suggested that functional syncytin may be limited in HEK293T cells, thus explaining the appearance of close-ended tubular connections between HEK293T cells (*Figure 6*).

We visualized HEK293T cells containing SBP-Cas9-GFP and Myc-streptavidin-CD63-mCherry co-cultured with MDA-MB-231 syncytin-2 knockdown cells containing the Fluc:Nluc:mCherry reporter plasmid and found close-ended tubular connections even after several hours of contact (*Figure 8A*, *Figure 8—video 1*). Further inspection by CLEM confirmed the failure of plasma membrane fusion in conditions of syncytin-2 knockdown in MDA-MB-231 cells (*Figure 8B–D*). In combining live-cell imaging and CLEM results, we found only one open-ended tubular connections in 120 tubule-cell contacts of co-cultures using MDA-MB-231 syncytin-2 knockdown cells. We conclude that syncytin-2 is required or rate-limiting for membrane fusion at the point of contact between a tubule and the other cell plasma membrane.

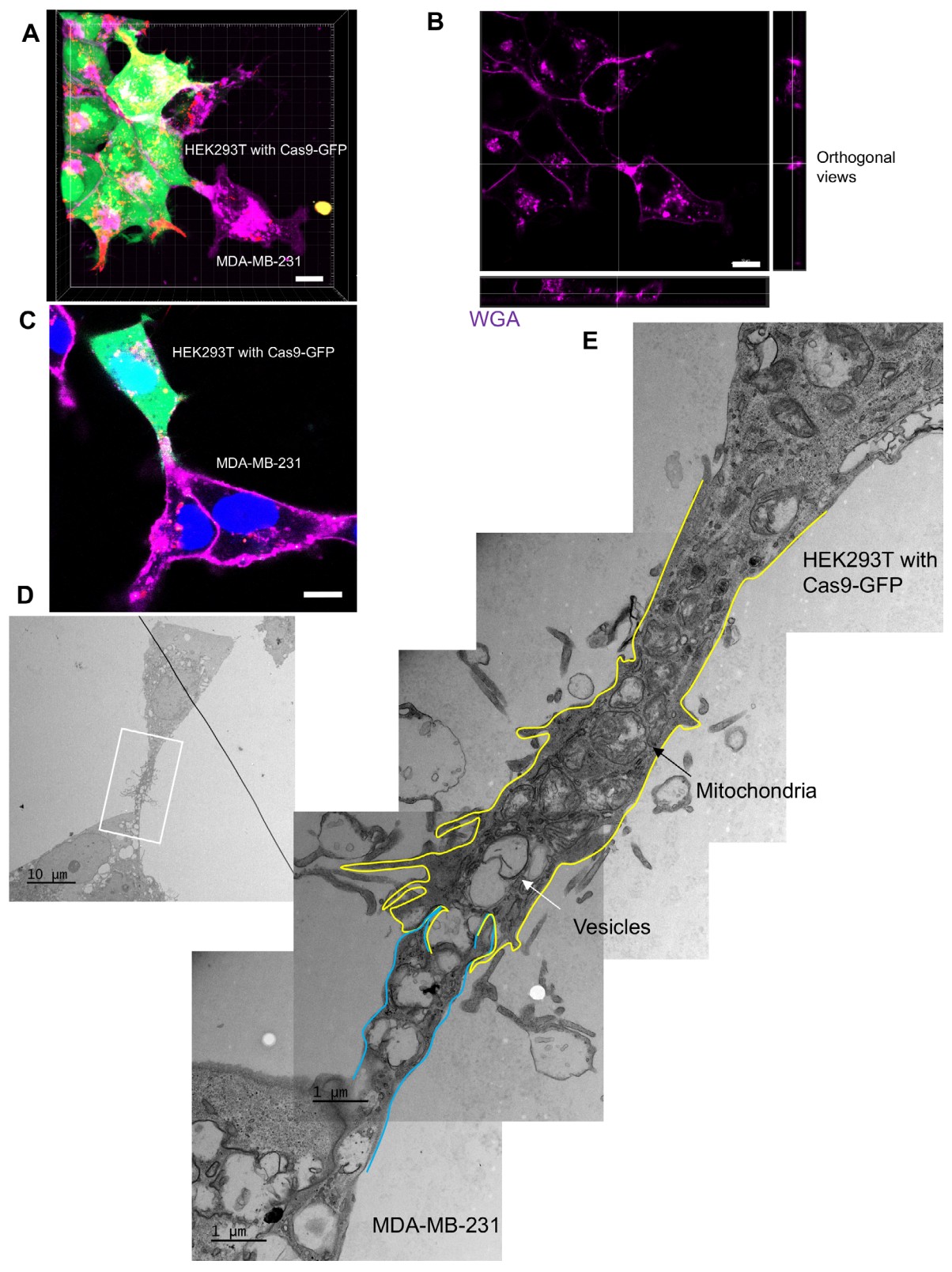

**Figure 5.** The ultrastructure of open-ended tubular connection confirmed by 3D tomography and correlative light and electron microscopy (CLEM). (**A, B**) 3D tomograph of open-ended tubular connection between HEK293T and MDA-MB-231. Z-stack images of the co-cultures were collected by confocal microscopy and analyzed using Imaris software. Donor cells: HEK293T with stable overexpression of SBP-Cas9-GFP/gRNA and Myc-streptavidin-CD63-mCherry (Str-CD63/SBP-Cas9/gRNA). Acceptor cell: MDA-MB-231. Green: SBP-Cas9-GFP; red: Myc-streptavidin-CD63-mCherry; purple: CF640R WGA

*Figure 5 continued on next page*

*Figure 5 continued*

conjugate. The top and lateral view of a contact site are shown in (**B**). Scale bar is 10 μm. (**C–E**) The ultrastructure of an open-ended tubular connection visualized by CLEM. (**C**) The open-ended tubular connection between HEK293T and MDA-MB-231 imaged by confocal microscopy. Green: SBP-Cas9-GFP; red: Myc-streptavidin-CD63-mCherry; purple: CF640R WGA conjugates; blue: Hoechst 33342. Scale bar is 10 μm. (**D**) The same area was imaged by transmission electron microscopy. Scale bar is 10 μm. (**E**) The area in white frame in (**D**) was examined and images were stacked. The plasma membrane of the membrane tube was traced manually with yellow line (from HEK293T) or blue line (from MDA-MB-231). Mitochondria and endosome-related vesicles were indicated by black and white arrows, respectively. Scale bar is 1 μm.

The online version of this article includes the following video(s) for figure 5:

**Figure 5—video 1.** Open-ended tubular connection visualized by 3D tomography.

https://elifesciences.org/articles/84391/figures#fig5video1

**Figure 5—video 2.** Open-ended tubular connection visualized by 3D tomography (WGA only channel).

https://elifesciences.org/articles/84391/figures#fig5video2

To further confirm the role of syncytins in membrane fusion at the site of cell-cell contact, we visualized the localization of syncytins by transient transfection of MDA-MB-231 cells with GFP-tagged forms of syncytin-A, -1, and -2. These results showed that most GFP-fusion proteins localized in a reticular pattern within transfected cells with some label detected at or near the cell surface (*Figure 8—figure supplement 1A–C*). We then applied cell surface biotinylation using an impermeable labeling reagent to test whether GFP-syncytins localized at cell surface. Syncytin-fusion proteins were immunoprecipitated with GFP antibody and SDS-PAGE blots were probed with horseradish peroxidase-conjugated streptavidin (streptavidin-HRP) and GFP antibody. HRP blots showed two species that migrated at the positions expected for full-length and furin processed forms of the syncytin fusion proteins (*Figure 8—figure supplement 1D*). In contrast, immunoprecipitated fractions blotted with GFP antibody revealed species that migrated at positions expected for the full-length fusion proteins. We interpret this to mean that the majority of molecules accumulated early in the secretory pathway prior to the point of cleavage by the furin protease in the trans-Golgi membrane and beyond.

Next, we visualized HEK293T cells containing SBP-Cas9-GFP and Myc-streptavidin-CD63-mCherry co-cultured with MDA-MB-231 cells containing syncytin 1-BFP (*Figure 8—figure supplement 1E*) or syncytin 2-BFP (*Figure 8—figure supplement 1F*). Syncytin-BFP fusions were found partially localized to or near the surface and at junctions between a tubule and the adjoining cell plasma membrane (*Figure 8—figure supplement 1E and F*). We conclude that under conditions of transient transfection at least some of all three syncytins may be detected at the cell surface and in proximity to sites of membrane fusion. The results of the GFP immunoblot of total proteins and biotinylation of those molecules trafficked to the cell surface suggest that some small fraction of the total fusion proteins experience proteolytic maturation.

We also examined the role of the syncytin-2 receptor, MFSD2A, established in studies on trophoblast cell fusion (*Esnault et al., 2008*). MFSD2A knockdown in HEK293T cells reduced functional Cas9 transfer by approximately five- to sixfold (*Figure 9A*). MFSD2A knockdown in MDA-MB-231 acceptor cells reduced Cas9 transfer approximately three- to fourfold (*Figure 9C and D*). These results suggested roles for MFSD2A in functional Cas9 transfer in both HEK293T and MDA-MB-231 cells. Given the above results reveal that cargos could be transferred among MDA-MB-231 cells (*Figure 2G*), the data in *Figure 9* suggest that syncytin and its receptor may also play roles in the intercellular transfer among MDA-MB-231 cells. Finally, we used IF and live-cell imaging and localized MFSD2A to the cell surface of normal and GFP-MFSD2A-transfected cells (*Figure 9E and F*).

## Discussion

In this study, we employed two reporter approaches designed to detect and quantify intercellular transfer of cargo proteins mediated by exosomes and cell contact-dependent connections. A Cas9-based dual-luciferase system and a trifluorescence split-GFP assay were used to measure Cas9-mediated genome editing and the formation of active GFP from fragments shared between cells in contact. We found that exosomes enriched in Cas9 were internalized by acceptor cells but largely failed to release Cas9 for editing purposes. The same was true for exosomes enriched in a fragment of GFP that failed to form active GFP in acceptor cells. In contrast, quite efficient transfer was seen in co-cultures of cells grown to near confluence. Transfer among cells in co-culture was dependent on

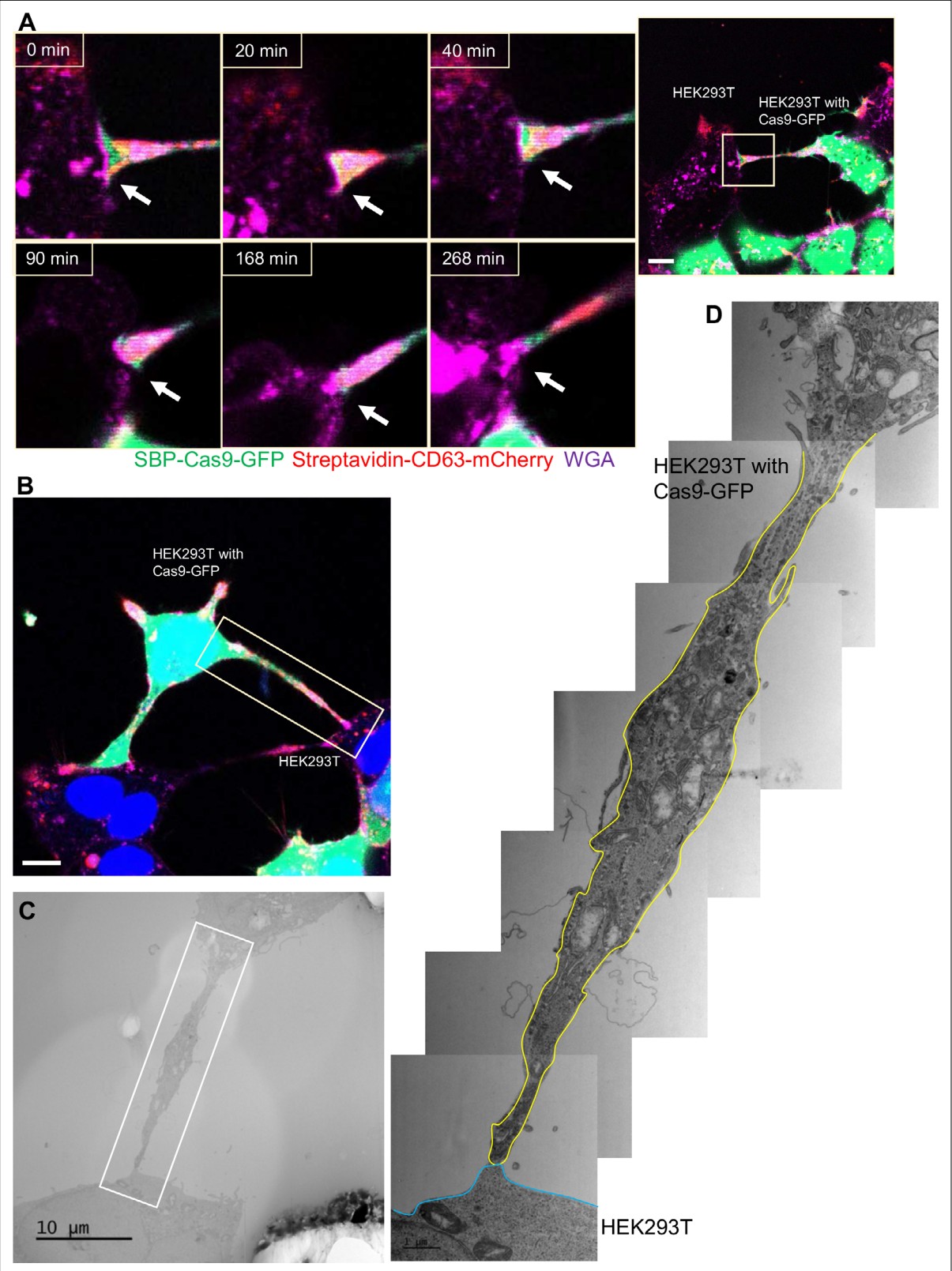

**Figure 6.** Close-ended tubular connection between HEK293T cells. (**A**) Visualization of close-ended membrane tube structure and time-lapse imaging of the co-culture. Donor cells: HEK293T with stable overexpression of SBP-Cas9-GFP/gRNA and Myc-streptavidin-CD63-mCherry (Str-CD63/SBP-Cas9/gRNA). Acceptor cell: HEK293T. Green: SBP-Cas9-GFP; red: Myc-streptavidin-CD63-mCherry; purple: CF640R WGA conjugate. Scale bar is 10 µm. The white arrows indicate a close-ended membrane tube. (**B–D**) Close-ended membrane tube ultrastructure visualized by correlative light and electron

*Figure 6 continued on next page*

*Figure 6 continued*

microscopy (CLEM). (**B**) The close-ended membrane tube between HEK293T cells was imaged by confocal microscopy. Green: SBP-Cas9-GFP; red: Myc-streptavidin-CD63-mCherry; purple: CF640R WGA conjugates; blue: Hoechst 33342. Scale bar is 10 µm. (**C**) The same area was imaged by transmission electron microscopy. Scale bar is 10 µm. (**D**) The area in white frame in (**C**) was examined and images were stacked. The plasma membrane of the membrane tube traced manually with a yellow (from HEK293T with Cas9-GFP) or blue line (from HEK293T WT). Scale bar is 1 µm.

The online version of this article includes the following video for figure 6:

**Figure 6—video 1.** Close-ended tunneling tubular connection formed between HEK293T cells.

https://elifesciences.org/articles/84391/figures#fig6video1

actin but was not blocked under conditions that arrested endocytosis. Imaging experiments revealed occasional intercellular tubular connections of a range of diameter that in certain donor-acceptor cell pairs appeared to be open-ended to allow cytoplasmic continuity (*Figures 4 and 9G*). As a donor cell line, HEK293T was quite efficient in cargo transfer and formed open-ended tubular connections to MDA-MB-231 acceptor cells. In contrast, the tubular connections between HEK293T cells were not open-ended and transferred little if any cargo. The transfer process appeared to consist of at least three stages: an initial tubular protrusion, possibly driven by F-actin, emanating from a donor or acceptor cell; contact by the tubule with an opposing cell surface; and membrane fusion in favorable circumstances, particularly when the acceptor cell was MDA-MB-231 or one of several other tumor cell lines. Tubules formed and made contact with opposing cells but did not fuse unless the endogenous membrane fusogen, syncytin (particularly syncytin-2), was expressed on the acceptor cell. The syncytin-2 receptor protein, MFSD2A, appeared to be required to promote fusion for the intercellular transfer.

Many reports have described the intercellular traffic of proteins, RNA, and organelles such as mitochondria and lysosomes mediated by tubular connections; however, the physiological function and molecular mechanism of this pathway have not been reported (*Pinto et al., 2020*; *Roehlecke and Schmidt, 2020*; *Yamashita et al., 2018*; *Dagar et al., 2021*). In contrast, exosomes are widely believed to mediate intercellular traffic of proteins and RNA, yet no reports have probed the mechanism of the membrane fusion event that must accompany the discharge of exosome content to the cytoplasm or nucleus of targeted cells.

We considered other pathways of traffic mediated by cell-cell contact that could explain our results. Trogocytosis is a means by which one cell nibbles another cell (*Joly and Hudrisier, 2003*). In this process, lymphocytes (B, T, and NK cells) extract surface molecules from antigen-presenting cells and express them on their own surface (*Joly and Hudrisier, 2003*; *Gutiérrez-Vázquez et al., 2013*). Previous studies reported that trogocytosis requires PI3K activation, which is blocked by such inhibitors as wortmannin (*Lis et al., 2010*; *Martínez-Martín et al., 2011*). We found that the PI3K inhibitors wortmannin and LY294002 did not block the transfer of Cas9 (*Figure 3A*). Alternatively, some cells in contact form an 'immunological synapse' that could promote the local traffic of exosomes possibly leading to selected delivery of cargo proteins and RNA (*Dustin, 2014*; *Grakoui et al., 1999*). However, we found intercellular traffic relatively unaffected by inhibitors of endocytosis, which would be required for the uptake of vesicles produced and consumed at an immunologic-like synapse.

A variety of other cell-cell adhesions, some of which involve tubular connections, have been reported. Many such connections remained close-ended such as in gap junctions, synaptic junctions, and cytonemes (*Yamashita et al., 2018*; *Kornberg and Roy, 2014*; *Ramírez-Weber and Kornberg, 1999*). We found large lateral surfaces form and remain stable at the junction of tubules and the cell surface under conditions where membrane fusion does not occur. These adhesions may require molecules employed for other stable cell-cell junctions, and open-ended membrane tubes such as TNTs and tumor microtubes (*Roehlecke and Schmidt, 2020*; *Yamashita et al., 2018*). The role of such stable junctions as a prelude to membrane fusion remains to be considered.

Fusion has been seen at other tubular junctions, particularly for the formation of open-ended contact with thin tubules called TNTs. TNTs range in diameter from 50 to 1000 nm and in length from a few to 100 µm (*Roehlecke and Schmidt, 2020*). Previous studies have reported the transfer of proteins, mRNA, and organelles such as mitochondria and lysosomes mediated by TNTs (*Pinto et al., 2020*). Tumor microtubes produced by cancer cells generate tubular connections of 1–2 µm in diameter, perhaps to allow the passage of larger organelles (*Roehlecke and Schmidt, 2020*; *Osswald et al., 2015*; *Latario et al., 2020*). Such membrane tubes were detected in tumors (*Osswald et al.,*

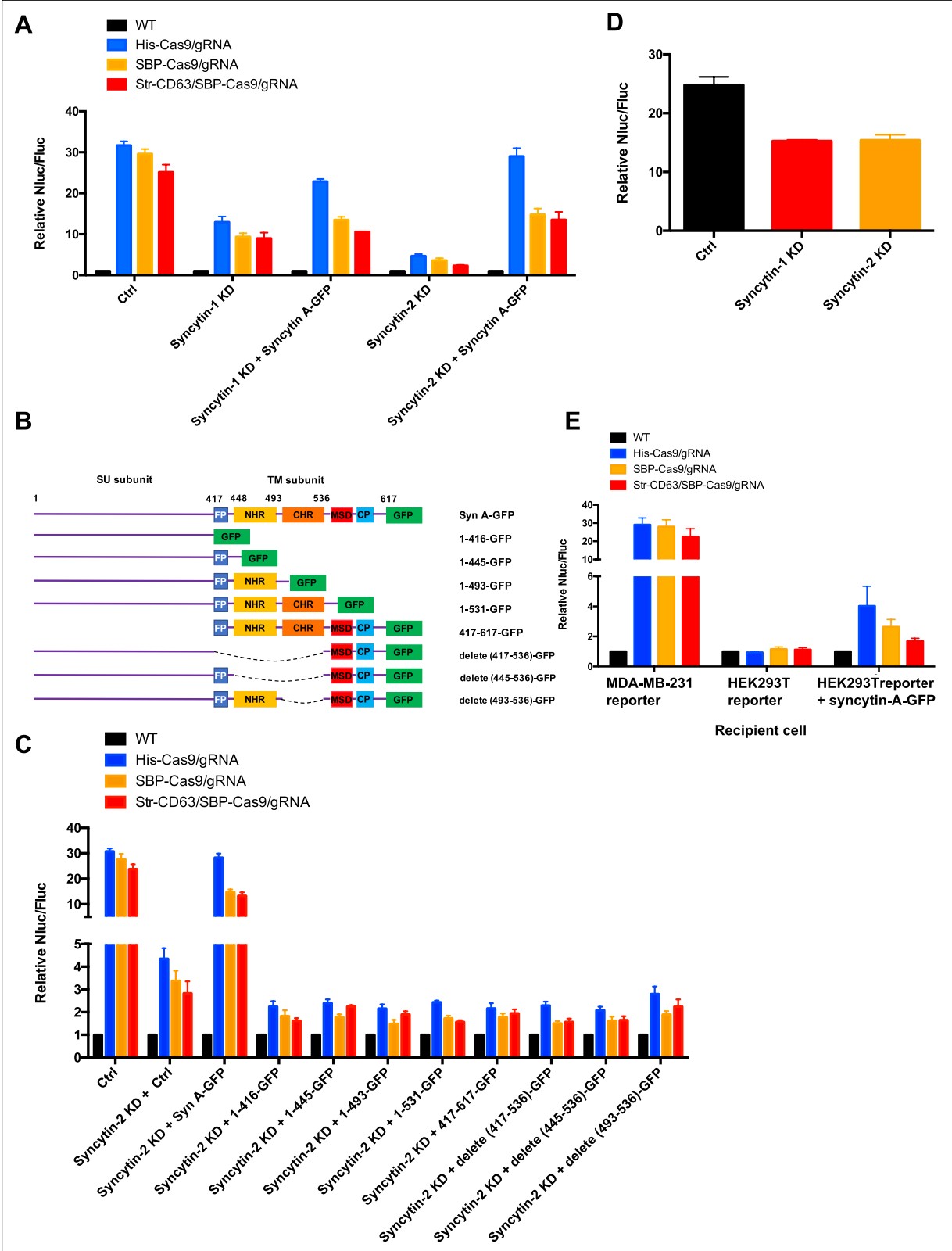

**Figure 7.** Syncytins in MDA-MB-231 regulate intercellular transfer of Cas9 protein. (**A**) Donor cells: HEK293T wild-type (WT) with stable overexpression of his tagged Cas9-GFP/gRNA (His-Cas9/gRNA), with stable overexpression of SBP-tagged Cas9-GFP/gRNA (SBP-Cas9/gRNA) or with stable overexpression of SBP-Cas9-GFP/gRNA and Myc-streptavidin-CD63-mCherry (Str-CD63/SBP-Cas9/gRNA). The recipient cells are MDA-MB-231 with reporter plasmid only (Ctrl), syncytin-1 knockdown (syncytin-1 KD), syncytin-1 knockdown, as well as expression of GFP-fused mouse syncytin A

*Figure 7 continued on next page*

*Figure 7 continued*

(syncytin-1 KD+syncytin A-GFP), syncytin-2 knockdown (syncytin-2 KD), or syncytin-2 knockdown as well as expression of GFP-fused mouse syncytin A (syncytin-2 KD+syncytin-A-GFP). Donor cells and acceptor cells were co-cultured for 3 days followed by quantitative assay of nanoluciferase and firefly luciferase. Nluc/Fluc assays were performed and normalized to an aliquot of co-cultured WT donor and reporter cells. Data represent mean ± SEM, n ≥ 3. (**B**) The architecture of mouse syncytin-A and schematic showing the truncations of syncytin-A. (**C**) Donor cells were the same as in (**A**). The recipient cells were MDA-MB-231 with reporter plasmid only (Ctrl), syncytin-2 knockdown (syncytin-2 KD+Ctrl), syncytin-2 knockdown, as well as expression of GFP-fused mouse syncytin-A (syncytin-2 KD+syncytin A-GFP), or syncytin-2 knockdown as well as expression of GFP-fused truncated syncytin-A (syncytin-2 KD+truncated syncytin A-GFP). Donor cells and recipient cells were co-cultured for 3 days. Nluc/Fluc assays were performed and normalized to an aliquot of co-cultured WT donor and reporter cells. Data represent mean ± SEM, n ≥ 3. (**D**) Donor cells: HEK293T with stable overexpression of SBP-tagged Cas9-GFP/gRNA (SBP-Cas9/gRNA) (CTRL), syncytin-1 knockdown, or syncytin-2 knockdown. The recipient cell line is MDA-MB-231 with reporter plasmid. The donor cells and recipient cells were co-cultured for 3 days. Nluc/Fluc assays were performed and normalized to an aliquot of co-cultured WT donor and reporter cells. Data represent mean ± SEM, n ≥ 3. (**E**) Donor cells were same as in (**A**). The recipient cells were MDA-MB-231 with reporter plasmid, HEK293T with reporter plasmid, or HEK293T with reporter plasmid as well as expression of GFP-fused mouse syncytin-A. The donor cells and recipient cells were co-cultured for 3 days. Nluc/Fluc assays were performed and normalized to an aliquot of co-cultured WT donor and reporter cells. Data represent mean ± SEM, n ≥ 3.

The online version of this article includes the following source data and figure supplement(s) for figure 7:

**Figure supplement 1.** Confirmation of syncytin expression and knockdown.

**Figure supplement 1—source data 1.** Uncropped Western blot images corresponding to *Figure 7—figure supplement 1A*.

**Figure supplement 1—source data 2.** Uncropped Western blot images corresponding to *Figure 7—figure supplement 1B*.

**Figure supplement 1—source data 3.** Uncropped Western blot images corresponding to *Figure 7—figure supplement 1D*.

**Figure supplement 1—source data 4.** Uncropped Western blot images corresponding to *Figure 7—figure supplement 1F*.

**Figure supplement 1—source data 5.** Uncropped Western blot images corresponding to *Figure 7—figure supplement 1G*.

**Figure supplement 1—source data 6.** Uncropped Western blot images corresponding to *Figure 7—figure supplement 1H*.

*2015*; *Lou et al., 2012*), possibly serving a role in tumor maintenance and progression. MDA-MB-231 cells transferred cytoplasmic cargo to other cancer cell lines, including U2OS, A549, and MCF5, as well as to other MDA-MB-231 cells (*Figure 2G*). The tubular connections we found range from 2 to 4 µm in diameter (*Figures 4–6*) and were not confined to tumor cells but included our standard donor cell line, HEK293T (*Figures 4–6* and *Figure 8*). We observed membrane tubules projecting from HEK293T cells connecting to MDA-MB-231 cells (*Figures 4 and 5*) and vice versa (*Figure 8*). Such structures from nontransformed cells could indicate a more normal role in cell and tissue physiology and at the border of a tumor as in the hijacking of mitochondria from immune cells (*Saha et al., 2022*).

At present, it is not possible to offer distinct roles for TNTs and thick tubules in the transfer of cytoplasmic cargo between cells. Our observations are consistent with either or both such structures engaged in the transfer of Cas9 and our other reporter proteins. Aside from the example of cancer cells and their interface in tumors, the role of TNTs and larger tubular connections in normal physiological functions remains to be explored.

Among the possible catalysts of open-ended connections, we considered the role of the one known mammalian viral-like fusogen, syncytin (syncytin-1 and syncytin-2). Syncytins are mammalian endogenous fusogens that facilitate trophoblast cell fusion in the early embryo (*Blaise et al., 2003*; *Mi et al., 2000*; *Grandi and Tramontano, 2018*). In this study, we used immunoblot to detect syncytins corresponding in SDS-PAGE mobility to precursor and/or furin-processed mature proteins expressed in several cell lines (*Figure 7—figure supplement 1*). Using CRISPRi, we found that knockdown of syncytins in MDA-MB-231 cells, but not in HEK293T cells, blocked transmission of Cas9 and the formation of open-ended tubular connections (*Figures 7 and 8*). Previous studies reported a number of other proteins involved in TNT formation, including M-Sec and the exocyst complex (*Hase et al., 2009*; *Kimura et al., 2016*), LST1 and RalA (*Schiller et al., 2013*), ERp29 (*Pergu et al., 2019*), S100A4 and RAGE (*Sun et al., 2012*), Myosin10 (*Gousset et al., 2013*), Rab8-Rab11-Rab35 (*Bhat et al., 2020*; *Zhu et al., 2018*), MICAL2PV (*Wang et al., 2021*), IRSp53 (*Prévost et al., 2015*; *Delage et al., 2016*), and others (*Dagar et al., 2021*). However, it is not clear in any of these cases if the protein is involved in tube formation, adhesion, or fusion. Nonetheless, it is likely that proteins, in addition to syncytin, such as the receptor proteins MFSD2A (for syncytin-2), ASCT2 (for syncytin-1), and Ly6e (for mouse syncytin-A), are involved specifically in fusion. The failure of endogenous syncytins to support fusion in HEK293T cells serving as an acceptor target could relate to unexplored aspects of intracellular traffic or regulation of fusogen activity. Although we found some increase in Cas9 traffic to HEK293T cells

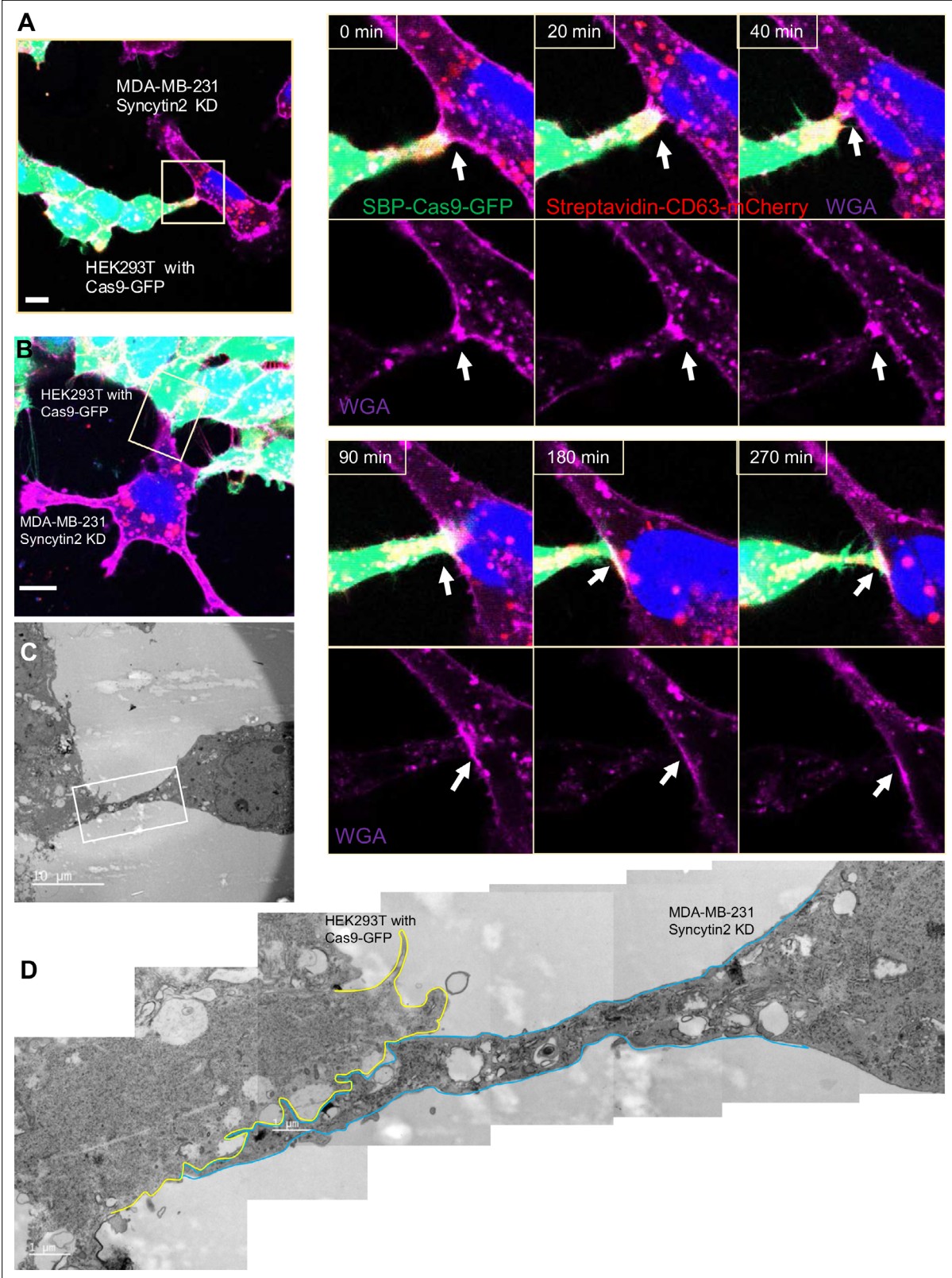

**Figure 8.** Syncytin-2 knockdown in MDA-MB-231 form close-ended tubular connection. (**A**) Visualization of close-ended membrane tube structure and time-lapse imaging of a co-culture of HEK293T with stable overexpression of Str-CD63/SBP-Cas9/gRNA with MDA-MB-231 Syncytin-2 knockdown cells containing Fluc:Nluc:mCherry. Green: SBP-Cas9-GFP; purple: CF640R WGA conjugates; blue: Hoechst 33342. Scale bar is 10 μm. The white arrows indicate a close-ended membrane tube. Of note, the red signal in MDA-MB-231 may be the mCherry signal from the reporter plasmid. (**B–D**) Close-

*Figure 8 continued on next page*

*Figure 8 continued*

ended membrane tube ultrastructure was visualized by correlative light and electron microscopy (CLEM). (**B**) The close-ended membrane tube between HEK293T cells and MDA-MB-231 with syncytin-2 knockdown was imaged by confocal microscopy. Green: SBP-Cas9-GFP; red: Myc-streptavidin-CD63-mCherry; purple: CF640R WGA conjugates; blue: Hoechst 33342. Scale bar is 10 µm. (**C**) The same area was imaged by transmission electron microscopy. Scale bar is 10 µm. (**D**) The area in white frame in (**C**) was examined and images were stacked. The plasma membrane of the membrane tube was traced manually with a yellow (from HEK293T with Cas9-GFP) or blue line (from MDA-MB-231 with Syncytin-2 knockdown). Scale bar is 1 µm.

The online version of this article includes the following video, source data, and figure supplement(s) for figure 8:

**Figure supplement 1.** Localization of syncytins at cell surface and sites of cell-cell contact.

**Figure supplement 1—source data 1.** Uncropped Western blot images corresponding to *Figure 8—figure supplement 1D*.

**Figure 8—video 1.** Syncytin-2 knockdown in MDA-MB-231 blocks plasma membrane fusion to form a close-ended tubular connection.
https://elifesciences.org/articles/84391/figures#fig8video1

ectopically expressing mouse syncytin-A (*Figure 7E*), full fusion activity may depend on unknown variables.

Inefficient sorting of functional syncytins or their receptors may explain the failure of exosomes isolated from HEK293T and MDA-MB-231 cells to deliver Cas9 for editing in the other target cell (*Figure 1*). Indeed, the requirement for an active fusogen incorporated into exosomes may explain other examples of failed delivery (*de Jong et al., 2020*; *Haimovich et al., 2017*; *Somiya and Kuroda, 2021*; *Albanese et al., 2021*). The question remains to explain the many observations that suggest functional delivery by exosomes. A closer look by quantitative measures may show that apparent success in exosome-mediated transfer is quite inefficient. A contrasting example may be in the activity of trophoblasts that may express active syncytins and their receptors at the cell surface and may likewise traffic extracellular vesicles within the developing placenta (*Tolosa et al., 2012*; *Uygur et al., 2019b*; *Vargas et al., 2014*). Lessons learned from trophoblasts may inform the rational design of functional exosomes useful in the delivery of heterologous protein, such as Cas9 protein, between cells.

# Materials and methods

**Key resources table**

| Reagent type (species) or resource | Designation | Source or reference | Identifiers | Additional information |
|---|---|---|---|---|
| Cell line (*Homo sapiens*) | HEK293T cells | Cell Culture Facility, UC Berkeley | N/A | |
| Cell line (*H. sapiens*) | U2OS cells | Cell Culture Facility, UC Berkeley | N/A | |
| Cell line (*H. sapiens*) | MDS-MB-231 cells | Cell Culture Facility, UC Berkeley | N/A | |
| Cell line (*H. sapiens*) | A549 cells | Cell Culture Facility, UC Berkeley | N/A | |
| Cell line (*H. sapiens*) | MCF7 cells | Cell Culture Facility, UC Berkeley | N/A | |
| Cell line (*H. sapiens*) | Hela cells | Cell Culture Facility, UC Berkeley | N/A | |
| Cell line (*H. sapiens*) | A431 cells | Cell Culture Facility, UC Berkeley | N/A | |
| Transfected construct (human) | siRNA to CLTC | QIAGEN | Hs_CLTC_10 FlexiTube siRNA, SI00299880 | Transfected construct (human) |
| Transfected construct (human) | siRNA to AP2B1 | QIAGEN | Hs_AP2B1_6 FlexiTube siRNA, SI02780085 | Transfected construct (human) |
| Transfected construct (human) | siRNA to CAV-1 | QIAGEN | Hs_CAV1_10 FlexiTube siRNA, SI00299642 | Transfected construct (human) |

*Continued on next page*

*Continued*

| Reagent type (species) or resource | Designation | Source or reference | Identifiers | Additional information |
|---|---|---|---|---|
| Transfected construct (human) | siRNA to FLOT2 | QIAGEN | Hs_FLOT2_5 FlexiTube siRNA, SI02781422 | Transfected construct (human) |
| Transfected construct (human) | siRNA to MFSD2A | QIAGEN | Hs_MFSD2A_1 FlexiTube siRNA, SI04137854 | Transfected construct (human) |
| Antibody | Anti-GFP (rabbit polyclonal) | Torrey Pines Biolabs | Cat# AB_10013661 | WB (1:1000) |
| Antibody | Anti-myc (mouse monoclonal) | Cell Signaling Technology | Cat# 2276s | WB (1:1000) |
| Antibody | Anti-syntenin (rabbit polyclonal) | Santa Cruz Biotechnology | Cat# sc-48742 | WB (1:200) |
| Antibody | Anti-TSG101 (mouse monoclonal) | GeneTex | Cat# GTX70255 | WB (1:1000) |
| Antibody | Anti-CD81 (mouse monoclonal) | Santa Cruz Biotechnology | Cat# sc-166029 | WB (1:200) |
| Antibody | Anti-CD9 (rabbit monoclonal) | Cell Signaling Technology | Cat# AB_2798139 | WB (1:1000) |
| Antibody | Anti-CD63 (rabbit monoclonal) | Abcam | Cat# ab134045 | IF (1:300) |
| Antibody | Anti-CLTC (mouse monoclonal) | BD Transduction Laboratories | Cat# AB_397866 | WB (1:1000) |
| Antibody | Anti-AP2B1 (rabbit polyclonal) | Proteintech | Cat# 15690-1-AP | WB (1:1000) |
| Antibody | Anti-CAV1 (rabbit polyclonal) | Proteintech | Cat# 16447-1-AP | WB (1:1000) |
| Antibody | Anti-FLOT2 (mouse monoclonal) | BD Transduction Laboratories | Cat# 610384 | WB (1:1000) |
| Antibody | Anti-actin (rabbit monoclonal) | Cell Signaling Technology | Cat# 8457S | WB (1:1000) |
| Antibody | Anti-tubulin (mouse monoclonal) | Abcam | Cat# ab7291 | WB (1:1000) |
| Antibody | Anti-Arp2 (rabbit polyclonal) | Proteintech | Cat# 10922-1-AP | WB (1:1000) |
| Antibody | Anti-Arp3 (rabbit polyclonal) | Proteintech | Cat# 13822-1-AP | WB (1:1000) |
| Antibody | Anti-syncytin 1 (rabbit polyclonal) | Thermo Fisher Scientific | Cat# BS-2962R | WB (1:1000) |
| Antibody | Anti-syncytin 2 (rabbit polyclonal) | Thermo Fisher Scientific | Cat# PA5-109694 | WB (1:1000) |
| Antibody | Anti-MFSD2A (rabbit polyclonal) | OriGene Technologies | Cat# TA351394 | WB (1:1000) IF (1:300) |
| Antibody | Anti-Cas9 (mouse monoclonal) | Novus Biologicals | Cat# NBP2-36440 | WB (1:1000) |
| Recombinant DNA reagent | pUCOE- EF1a-dCas9-BFP-KRAB (plasmid) | Dr. Jonathan Weissman (Whitehead Institute, MIT) | | Express dCas9-BFP-KRAB in cells for CRISPRi |
| Recombinant DNA reagent | pMyc-streptavidin-CD63-mCherry (plasmid) | This study | | Express Myc-streptavidin-CD63-mCherry in cells |
| Recombinant DNA reagent | pSBP-flag-Cas9-GFP (plasmid) | This study | | Express SBP-flag-Cas9-GFP in cells |
| Recombinant DNA reagent | pHis-flag-Cas9-GFP (plasmid) | This study | | His-flag-Cas9-GFP in cells |
| Recombinant DNA reagent | pFluc-cas9 target-Nluc-IRES-mCherry (plasmid) | This study | | Reporter for Cas9 editing |
| Recombinant DNA reagent | pFluc-cas9 target-Nluc1-Nluc2-IRES-mCherry (plasmid) | This study | | Reporter for Cas9 editing |
| Recombinant DNA reagent | pRluc-FLXXUC-IRES-mCherry (plasmid) | This study | | Reporter for Cas9 editing |
| Recombinant DNA reagent | pLKO.1-shactin (plasmid) | This study | | Lentiviral construct to transfect and express the shRNA |

*Continued on next page*

*Continued*

| Reagent type (species) or resource | Designation | Source or reference | Identifiers | Additional information |
|---|---|---|---|---|
| Recombinant DNA reagent | pLKO.1-sharp2 (plasmid) | This study | | Lentiviral construct to transfect and express the shRNA |
| Recombinant DNA reagent | pLKO.1-sharp3 (plasmid) | This study | | Lentiviral construct to transfect and express the shRNA |
| Recombinant DNA reagent | pCD63-7GFP11-IRES-CFP (plasmid) | This study | | Express CD63-7GFP11 and CFP in cells |
| Recombinant DNA reagent | pGFP10-IRES-mCherry (plasmid) | This study | | Express GFP10 and mCherry in cells |
| Recombinant DNA reagent | pU6_sgRNA_CAG_puroR (plasmid) | Addgene | | Express sgRNA for CRISPRi |
| Recombinant DNA reagent | pSyncytin A-GFP (plasmid) | This study | | Express Syncytin A-GFP in cells |
| Recombinant DNA reagent | pSyncytin A (1-416)-GFP (plasmid) | This study | | Express Syncytin A (1-416)-GFP in cells |
| Recombinant DNA reagent | pSyncytin A (1-445)-GFP (plasmid) | This study | | Express Syncytin A (1-445)-GFP in cells |
| Recombinant DNA reagent | pSyncytin A (1-493)-GFP (plasmid) | This study | | Express Syncytin A (1-493)-GFP in cells |
| Recombinant DNA reagent | pSyncytin A (1-531)-GFP (plasmid) | This study | | Express Syncytin A (1-531)-GFP in cells |
| Recombinant DNA reagent | pSyncytin A (417-617)-GFP (plasmid) | This study | | Express Syncytin A (417-617)-GFP in cells |
| Recombinant DNA reagent | pSyncytin A delete (417-536)-GFP (plasmid) | This study | | Express Syncytin A delete (417-536)-GFP in cells |
| Recombinant DNA reagent | pSyncytin A delete (445-536)-GFP (plasmid) | This study | | Express Syncytin A delete (445-536)-GFP in cells |
| Recombinant DNA reagent | pSyncytin A delete (493-536)-GFP (plasmid) | This study | | Express Syncytin A delete (493-536)-GFP in cells |
| Recombinant DNA reagent | pSyncytin-1-BFP (plasmid) | This study | | Syncytin-1-BFP in cells |
| Recombinant DNA reagent | pSyncytin-2-BFP (plasmid) | This study | | Syncytin-2-BFP in cells |
| Recombinant DNA reagent | pLenti-Syncytin-1-GFP11 (plasmid) | This study | | Express Syncytin-1-GFP11 in cells |
| Recombinant DNA reagent | pLenti-Syncytin-2-GFP11 (plasmid) | This study | | Express Syncytin-2-GFP11 in cells |
| Recombinant DNA reagent | pSyncytin-1-GFP (plasmid) | This study | | Express Syncytin-1-GFP in cells |
| Recombinant DNA reagent | pSyncytin-2-GFP (plasmid) | This study | | Express Syncytin-2-GFP in cells |
| Recombinant DNA reagent | pMFSD2A-GFP (plasmid) | This study | | Express MFSD2A-GFP in cells |
| Chemical compound, drug | Biotin | Sigma | Cat# B4639 | |
| Chemical compound, drug | Proteinase K | Sigma | Cat# P2308 | |

*Continued on next page*

*Continued*

| Reagent type (species) or resource | Designation | Source or reference | Identifiers | Additional information |
|---|---|---|---|---|
| Chemical compound, drug | Micrococcal Nuclease | NEB | Cat# M0247S | |
| Chemical compound, drug | ANTI-FLAG M2 Affinity Gel | Sigma | Cat# A2220 | |
| Chemical compound, drug | SuperSignal West Pico PLUS | Thermo Fisher Scientific | Cat# PI34580 | |
| Chemical compound, drug | SuperSignal West Femto | Thermo Fisher Scientific | Cat# PI34096 | |
| Chemical compound, drug | TGIRT-III Enzyme | InGex | Matthew et al., PNAS (2017) | |
| Commercial assay or kit | Direct-zol RNA Miniprep Plus Kit | Zymo research | Cat# R2072 | |
| Commercial assay or kit | mirVana miRNA isolation kit | Thermo Fisher Scientific | Cat# AM1560 | |
| Commercial assay or kit | Nano-Glo Luciferase Assay | Promega | Cat# N1150 | |
| Commercial assay or kit | Luciferase Assay System | Promega | Cat# E4550 | |
| Commercial assay or kit | Luciferase reporter assay | Promega | Cat# E1910 | |
| Chemical compound, drug | PowerUp SYBR Green Master Mix | Thermo Fisher Scientific | Cat# A25741 | |
| Chemical compound, drug | Chlorpromazine | Sigma | Cat# C8138 | 5 μg/ml |
| Chemical compound, drug | LY294002 | Sigma | Cat# L9908 | 10 μM |
| Chemical compound, drug | Wortmannin | Sigma | Cat# W1628 | 1 μM |
| Chemical compound, drug | Latrunculin A | Enzo | Cat# BML-T119-0100 | 40, 80, 200 nM |
| Chemical compound, drug | Latrunculin B | Sigma | Cat# 428020 | 1, 2.5, 5 μM |
| Chemical compound, drug | SMIFH2 | Sigma | Cat# 344092 | 10, 25 μM |
| Chemical compound, drug | Hoechst 33342 | Thermo Fisher Scientific | Cat# H3570 | |
| Chemical compound, drug | Phrodo Red Transferrin Conjugate | Thermo Fisher Scientific | Cat# P35376 | |
| Chemical compound, drug | Phrodo Green Zymosan Bioparticles | Thermo Fisher Scientific | Cat# P35365 | |
| Chemical compound, drug | Molecular Probes pHrodo Green Dextran | Thermo Fisher Scientific | Cat# P35368 | |
| Chemical compound, drug | Sulfo-NHS-LC-Biotin | Thermo Fisher Scientific | Cat# 21335 | |
| Chemical compound, drug | GFP-Trap agarose beads | ChromoTek | Cat# gta-20 | |
| Chemical compound, drug | Streptavidin-HRP Conjugate | Thermo Fisher Scientific | Cat# SA10001 | |
| Chemical compound, drug | CF640R Wheat Germ Agglutinin (WGA) Conjugates | Biotium | Cat# 29026-1 | |
| Chemical compound, drug | Lipofectamine 2000 Transfection Reagent | Thermo Fisher Scientific | Cat# 11668019 | |

*Continued on next page*

*Continued*

| Reagent type (species) or resource | Designation | Source or reference | Identifiers | Additional information |
|---|---|---|---|---|
| Chemical compound, drug | Lipofectamine RNAiMAX Transfection Reagent | Thermo Fisher Scientific | Cat# 13778150 | |
| Other | Glass-bottom dish | Thermo Fisher Scientific | Cat# NC0699576 (Cellvis D35-20-1-N) | For confocal microscopy experiments |
| Other | Gridded glass-bottom dish | Thermo Fisher Scientific | Cat# NC1144968 | For CLEM experiments |
| Software, algorithm | Fiji | NIH | | https://fiji.sc/ |
| Software, algorithm | GraphPad Prism | GraphPad | | https://www.graphpad.com |
| Software, algorithm | Imaris | Oxford Instruments | | https://imaris.oxinst.com/ |

## Cell lines and cell culture

Human HEK293T cells and cancer cell lines, including U2OS, MDS-MB-231, A549, MCF7, Hela, and A431, were from the UC Berkeley Cell Culture Facility and were confirmed by short tandem repeat profiling (STR) and tested negative for mycoplasma contamination. These cells were cultured in DMEM with 10% FBS with 100 units/ml penicillin and 100 units/ml streptomycin. For exosome production, we seeded cells to ~10% confluency in 150 mm tissue culture dishes (Corning, Corning, NY) containing 20 ml of growth medium and then grown to 80% confluency (~48 hr). Cells used for exosome production were incubated in exosome-free medium produced by ultracentrifugation at 40,000 rpm for 24 hr using a Ti-45 rotor (Beckman Coulter, Brea, CA) in a LE-80 ultracentrifuge (Beckman Coulter). For co-culture experiments with cell-cell contact, unless otherwise noted, we used a ratio of donor cells to recipient cells of 10:1. Cells were co-cultured for 3–6 days until near confluence. Co-cultures were assayed for luciferase activity or were analyzed by flow cytometry. For co-culture experiments using a transwell dish, we seeded donor cell lines in the top layer and reporter cells in the bottom layer followed by incubation for 6 days. Inhibitors with indicated concentrations were incubated in the co-cultures for 3–6 days.

For the stable cell lines, we transfected cells with lentivirus and then single clones were selected via antibiotics or sorted by flow cytometry. Plasmids, including His-Flag-Cas9-GFP, SBP-Flag-Cas9-GFP, Myc-streptavidin-CD63-mCherry, or the reporter plasmids, were transfected into HEK293T cells, together with psPAX2 and pMD2.G, for lentivirus packaging.

## siRNA and shRNA transfection

siRNAs were purchased from QIAGEN. Cells were transfected using Lipofectamine RNAiMAX Transfection Reagent according to the manufacturer's instructions. For shRNAs, the sequences were designed using online data available from the Broad Institute (https://portals.broadinstitute.org/gpp/public/) and synthesized by IDT and inserted into the pLKO.1 plasmid. Plasmids were transfected into HEK293T cells, together with psPAX2 and pMD2.G, for lentivirus packaging.

## CRISPR interference

MDA-MB-231 cells with reporter plasmid and HEK293T cells expressing dCas9-KRAB, as in the previous study (*Gilbert et al., 2013*), were generated using lentivirus. A modified version of the transfer plasmid, UCOE-EF1α-dCas9-BFP-KRAB, was kindly provided by Jonathan Weissman (Whitehead Institute, MIT). Cells were sorted using the single-cell mode for BFP signal post transduction. Sequences for gRNAs targeting the promoter of the genes of interest were extracted from the previous study (*Horlbeck et al., 2016*). gRNAs were cloned in plasmid pU6-sgRNA EF1Alpha-puro-T2A-BFP (*Gilbert et al., 2014*) and plasmid #60955 obtained from Addgene. The three top gRNAs from the V.2 library (*Horlbeck et al., 2016*) were chosen per gene of interest. Lentiviruses with the gRNAs targeting the genes of interest were used to transduce the parental cells. Three days post transduction, cells were sorted using a single-cell mode. Knockdown efficiencies were evaluated by immunoblot.

To avoid the effect of CRISPRi on a transfer assay that depended on the expression of Cas9, we titrated the expression of dCas9 in MDA-MB-231 reporter cells. Three representative clones were selected for high, medium, and low levels expression of dCas9. The high level of dCas9 expression was around threefold of that in medium-level clone and around tenfold of that in low-level clone.

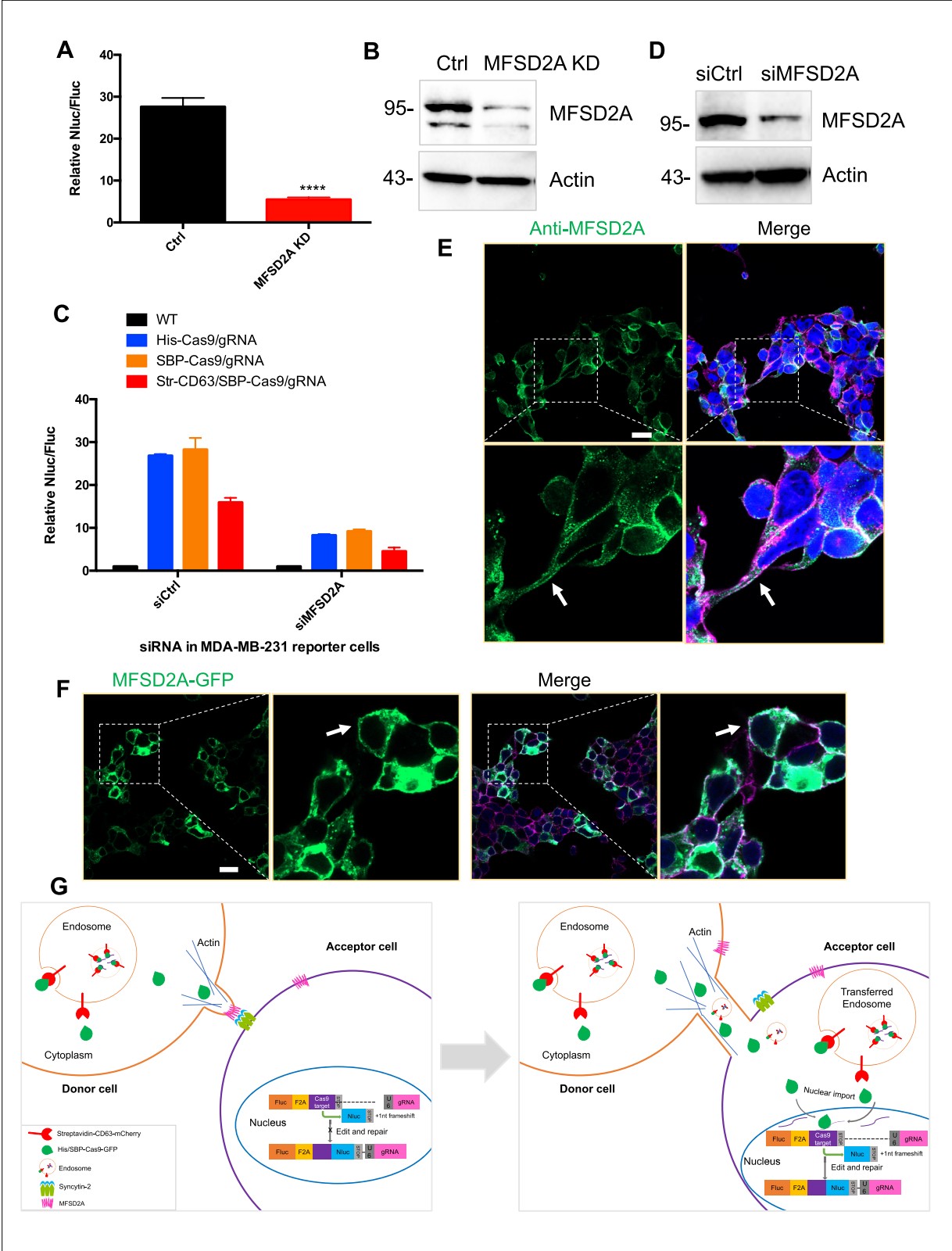

**Figure 9.** MFSD2A knockdown in in both HEK293T and MDA-MB-231 cells reduced intercellular transfer of Cas9. (**A**) Donor cells: HEK293T with stable overexpression of SBP-tagged Cas9-GFP/gRNA (SBP-Cas9/gRNA) (Ctrl), or MFSD2A knockdown. The recipient cell line is MDA-MB-231 with reporter plasmid. The donor cells and recipient cells were co-cultured for 3 days. Nluc/Fluc was measured and the data normalized to WT donor cells. Data represent mean ± SEM, n ≥ 3. ****p<0.0001, one-way ANOVA. (**B**) MFSD2A was knocked down in HEK293T with stable overexpression of SBP-tagged

*Figure 9 continued*

Cas9-GFP/gRNA (SBP-Cas9/gRNA). (**C**) Donor cells: HEK293T wild-type (WT) with stable overexpression of his-tagged Cas9-GFP/gRNA (His-Cas9/gRNA), with stable overexpression of SBP-tagged Cas9-GFP/gRNA (SBP-Cas9/gRNA) or with stable overexpression of SBP-Cas9-GFP/gRNA and myc-streptavidin-CD63-mCherry (Str-CD63/SBP-Cas9/gRNA). The recipient cells are MDA-MB-231 with reporter plasmid (siCtrl), or MFSD2A knockdown (siMFSD2A). Donor cells and recipient cells were co-cultured for 3 days. Nluc/Fluc assays were performed and normalized to an aliquot of co-cultured WT donor and reporter cells. Data represent mean ± SEM, n ≥ 3. (**D**) MFSD2A was knocked- own in MDA-MB-231 with reporter plasmid. (**E**) MFSD2A localization in HEK293T cells was detected by immunofluorescence using anti-MFSD2A antibody (green). The cells were stained by Hoechst 33342 and a CF640R-WGA conjugate. White arrows indicate the localization of MFSD2A at or near the plasma membrane. Scale bar is 20 µm. (**F**) MFSD2A-GFP was transfected in HEK293T cells. Cells were fixed and stained with a CF640R-WGA conjugate. White arrows indicate the localization of MFSD2A at or near the plasma membrane. Scale bar is 20 µm. (**G**) Model: donor and recipient cells form actin-based tubular protrusions projecting from donor cells (or both of donor and recipient cells) (left). Tubules adhere to recipient cell surface. The human endogenous fusogen, syncytin (especially syncytin-2), expressed on the acceptor cell surface then interacts with its receptor, MFSD2A, on the donor cell surface to facilitate plasma membrane fusion forming an open-ended tubular connection. Cargo, including endosomes and mitochondria, as well as Cas9 protein (free or bound to endosomes) and presumably other proteins and RNA, transfer from donor cells to recipient cells (right). Cytoplasmic Cas9 protein enters the nucleus inducing genome editing and Nluc expression (right). Of note, in cells such as MDA-MB-231, both syncytin-2 and MFSD2A are functional.

The online version of this article includes the following source data for figure 9:

**Source data 1.** Uncropped Western blot images corresponding to *Figure 9B*.

**Source data 2.** Uncropped Western blot images corresponding to *Figure 9D*.

We found that a low or medium level of expression of dCas9 did not itself significantly affect the detection of functional Cas9 transfer from HEK293T to MDA-MB-231 cells. We used a medium level of expression of dCas9 in the syncytin-1 knockdown clone, and a low level of dCas9 in the syncytin-2 knockdown clone, respectively (*Figure 7—figure supplement 1*).

## Immunoprecipitation

Cells expressing both SBP-Flag-Cas9-GFP and Myc-streptavidin-CD63-mCherry were cultured with or without biotin at indicated concentrations for 24 hr and then the cells were lysed with lysis buffer (50 mM Tris-HCl, pH7.4, 150 mM NaCl, 1 mM EDTA, 1% NP40, 1 mM PMSF) for 30 min at 4°. After centrifugation at 15,000 × *g* for 15 min, the supernatant fraction was collected and incubated with anti-Flag agarose beads (Sigma) for 2–3 hr at room temperature (RT) according to the manufacturer's instructions. After washing four times, beads were mixed with sample buffer, heated to 95° for 5 min, and aliquots were evaluated by SDS-PAGE and immunoblot.

## Immunoblotting

Proteins from cells and exosomes were extracted using 1X Laemmli sample buffer followed by heating at 95°C. Proteins were separated on 4–20% acrylamide Tris-Glycine gradient gels (Life Technologies), transferred to polyvinylidene difluoride membranes (EMD Millipore, Darmstadt, Germany), blocked with 5% bovine serum albumin in TBST, and incubated overnight with primary antibodies. Blots were then washed with TBST, incubated with anti-rabbit or anti-mouse secondary antibodies (GE Healthcare Life Sciences, Pittsbugh, PA), and detected with ECL-2 reagent (Thermo Fisher Scientific). Primary antibodies used in this study are listed in the Key Resources Table.

## Exosome purification

The protocol is described in detail in a previous study (*Shurtleff et al., 2016*). In brief, conditioned medium from donor cells was collected and centrifuged to remove cells and debris in a Sorvall R6+ centrifuge (Thermo Fisher Scientific) at 1000 × *g* for 20 min followed by further clarification at 10,000 × *g* for 30 min in 500 ml vessels using a fixed angle FIBERlite F14–6X500y rotor (Thermo Fisher Scientific). The supernatant fraction was then centrifuged at 29,500 rpm for 1.5 hr in Beckman SW-32 rotors. The pellet material was resuspended by adding 500 µl of phosphate-buffered saline, pH 7.4 (PBS), to each tube followed by trituration using a large bore pipette over a 30 min period at 4°C. The resuspended material was then diluted in 60% sucrose buffer (20 mM Tris-HCl, pH 7.4, 137 mM NaCl) and mixed evenly followed by layers of 40% and 10% sucrose buffer. Step gradient tubes were then centrifuged at ~150,000 × *g* (38,500 rpm) for 16 hr in a Beckman SW-55 rotor. The 10/40% interface was harvested, diluted 1:5 with PBS (pH 7.4), and centrifuged at ~150,000 × *g* (38,500 rpm) for 1 hr in an SW-55 rotor. Final pellet fractions were resuspended in PBS. Exosomes were quantified using

a Nanosight particle-tracking LM10 instrument (Malvern, UK) as in a previous study (*Temoche-Diaz et al., 2019*). RNA from the exosome samples was extracted using Direct-Zol RNA mini-prep (Zymo Research, Irvine, CA) and protein was extracted in 1X Laemmli sample buffer.

## Real-time-qPCR

RNA was extracted using either Direct-zol RNA Miniprep kits (Zymo Research) or a mirVana miRNA isolation kit (Thermo Fisher Scientific) according to the manufacturer's instructions. We used total RNA from cells or exosomes for normalization as there is no well-accepted control transcript for exosomes. Total RNA from cells was quantified by nanodrop, and total RNA from exosomes was quantified using an RNA bioanalyzer (Agilent). Typically, 10 ng total RNA from cells and exosomes was reverse transcribed using TGIRT-III Enzyme and the gRNA primer 5'-ACTCGGTGCCACTTTTTCAA GTT-3'. PowerUp SYBR Green Master Mix was used for real-time PCR, and reactions were performed on an ABI-7900 real-time PCR system (Life Technologies). For all RT-PCR reactions, the results are presented as mean cycle threshold (Ct) values of three independent technical replicates. Samples with a Ct value >40 were regarded as negative.

## Proteinase protection assay

Exosomes were purified as above and incubated with 10 ug/ml proteinase K in buffer (20 mM Tris-HCl, pH 7.4, 137 mM NaCl) with or without 1% Triton X-100 on ice for 20 min followed by 5 mM PMSF to inactive proteinase K. Sample buffer was added, heated to 95°C, and proteins were evaluated by SDS-PAGE and immunoblot.

## Nuclease protection assay

Exosomes were purified as above and incubated with 400 U/ul micrococcal nuclease in the reaction buffer (NEB) with or without 1% Triton X-100 at 37°C for 15 min followed by 0.2 M EDTA to inactive nuclease. The samples were analyzed using real-time-qPCR as above.

## Immunofluorescence

Engineered exosomes were purified as above and incubated with U2OS cell line (the ratio of exosome to cell is around 100,000:1) for 16 hr. Cells were washed with PBS 3×, fixed in 4% PFA, and permeabilized with 0.1% Triton-100. After blocking, anti-GFP and anti-CD63 antibodies were incubated with fixed, permeabilized cells at 4°C overnight and then with Alexa Fluor 488- and Alexa Fluor 560-conjugated secondary antibodies for 1 hr at RT. Cells were visualized with an LSM980 Airyscan confocal microscope.

HEK293T cells were washed with PBS 3×, fixed in 4% PFA. After blocking, anti-MFSD2A antibody was incubated with fixed cells at 4°C overnight and then with Alexa Fluor 488-conjugated secondary antibodies for 1 hr at RT. Cells were visualized with an LSM980 Airyscan confocal microscope.

## Luciferase assays

Two dual-luciferase systems were constructed: firefly luciferase (Fluc) and renilla luciferase (Rluc) and nano-luciferase (Nluc) and Fluc. The gene sequences and the assays for luminescence (Nano-Glo Luciferase Assay, Luciferase Assay System, and Luciferase reporter) followed Promega instructions. Luciferase activity was measured using a Promega Glowmax 20/20 luminometer (Promega, Madison, WI) with a signal collection integration time of 2 s. Nluc/Fluc assays were performed and normalized to an aliquot of co-cultured WT donor and reporter cells. Data represent mean ± SEM, n ≥ 3.

## Imaging

For confirmation of endocytosis inhibitors and endocytosis protein knockdown, MDA-MB-231 cells were incubated with indicated inhibitors or were transfected with siRNAs for 48 hr. Cells were washed 2× and incubated with pHrodo red transferrin conjugate and Hoechst 33342 for 20 min, washed 4× and observed with an Airyscan LSM900 confocal microscope. The integrated intensity per cell was quantified by ImageJ. n = 200 cells were captured. Three independent experiments were performed. Data represent mean ± SEM.

For live-cell imaging of co-cultures, cells were seeded in a glass-bottom dish. Donor cells: HEK293T with stable overexpression of SBP-Cas9-GFP/gRNA and Myc-streptavidin-CD63-mCherry (Str-CD63/

SBP-Cas9/gRNA), or MFSD2A depletion. Acceptor cells: MDA-MB-231 WT, syncytin-2 depletion, or HEK293T WT cells. Co-cultures were stained using CF640R-tagged fluorescent WGA conjugates and Hoechst 33342. Images were captured every 2 min for several hours by an Airyscan LSM900 confocal microscope.

For CLEM experiments, cells were co-cultured in a dish with a gridded coverslip. Cells of interest were captured by an Airyscan LSM900 confocal microscope, marked, and then fixed immediately using 2% glutaraldehyde and 2% paraformaldehyde for 1 hr at RT. Fixed cells were washed 3× with PBS buffer and post fixed with 1% osmium containing 1.6% potassium ferrocyanide for 30 min at RT. All samples were then dehydrated in a graded series of ethanol (30, 50, 70, 95, 100, and 100%) for 7 min each. Samples were infiltrated with and embedded in resin. After polymerizing overnight at 60°C, the coverslips were removed from the bottom of the live-cell dishes. Next, 70-nm-thick ultrathin sections were cut using a diamond knife and then picked up with Formvar-coated copper grids. Sections were double-stained with uranyl acetate and lead citrate. After air drying, samples were examined with a FEI Tecnai 12 transmission electron microscope. All the EM experiments were performed in the Electron Microscope Laboratory at UC Berkeley.

## Flow cytometry

Cells or co-cultures were analyzed or sorted using flow cytometry. FSC-A and FSC-H were used to gate for single particles (singlets), which were used for further analysis. Gating of each fluorescent channel was determined by comparing a control sample without any fluorescence labeling and a control that was labeled in a single channel. Data were collected on a BD Bioscience LSR Fortessa (Becton Dickinson, Franklin Lakes, NJ) and analyzed by FlowJo X software and Flowing software. Instruments and software were provided by the LKS flow core facility at UC Berkeley.

## Detection of cell-surface syncytins

MDA-MB-231 cells transiently expressing GFP-tagged syncytin A, -1, or -2 were collected and incubated with 2 mM sulfo-NHS-LC-biotin according to the manufacturer's instructions. The reaction was quenched with 100 mM glycine in PBS. Cells were centrifuged and mixed with the lysis buffer (10 mM Tris/Cl pH 7.5, 150 mM NaCl, 0.5 mM EDTA, 0.5% Nonidet P40) and total GFP-tagged syncytins were immunoprecipitated using GFP-TRAP agarose according to the manufacturer's instructions. Isolated GFP-tagged syncytins were processed for SDS–PAGE. Total immunoprecipitated fusion proteins were detected by GFP immunoblot and biotinylated GFP-tagged syncytins were detected using horseradish peroxidase (HRP)-conjugated streptavidin.

## Statistical analysis

Statistical analysis was performed in GraphPad Prism. All data were obtained from more than three independent experiments. Data represent mean ± SEM. Differences were assessed with two-tailed Student's $t$-test or one-way ANOVA. p-Values ≤ 0.05 were considered significant. *p<0.05, **p<0.01, ***p<0.001, ****p<0.0001.

## Acknowledgements

We dedicate this work to the memory of Robert Lesch, our lab manager for the past several decades who was tragically taken from us in an accident in 2021. We thank Dr. Elizabeth Chen for her suggestions about syncytin and Drs David Drubin, Criss Hartzell, Ross Wilson, and Mr. Armeen Mozaffari for helpful discussions and suggestions. We thank Drs Xuedong Liu and Xiaojuan Zhang for the gift of cDNAs of codon-optimized syncytin-1, and -2. We thank Liang Ma and Evan Luke Carpenter for reading and editing the manuscript. We also thank the staff at the UC Berkeley shared facilities, the Electron Microscopy Laboratory (Danielle Jorgens and Reena Zalpuri III), the cell culture facility (Alison Killilea), Biological imaging facility (Steven Ruzin), the Flow Cytometry Facility (Hector Nolla, Alma Nuguid, and Kartoosh Heydari) and the DNA sequencing facility. CZ is supported as an Associate of the HHMI and UC Berkeley. RS is an Investigator of the HHMI, a senior fellow of the UC Berkeley Miller Institute of Science and the Scientific Director of Aligning Science Across Parkinson's Disease (ASAP). The work was supported by the HHMI and by funds from the UC Berkeley IGI and the Sergey Brin Family Foundation.

## Additional information

### Competing interests
Randy Schekman: Reviewing editor, *eLife*. The other author declares that no competing interests exist.

### Funding

| Funder | Grant reference number | Author |
|---|---|---|
| Howard Hughes Medical Institute | | Randy Schekman |
| Innovative Genomics Institute, UC Berkeley | | Randy Schekman |
| Sergey Brin Family Foundation | | Randy Schekman |

The funders had no role in study design, data collection and interpretation, or the decision to submit the work for publication.

### Author contributions
Congyan Zhang, Conceptualization, Validation, Investigation, Visualization, Methodology, Writing - original draft, Writing - review and editing; Randy Schekman, Conceptualization, Resources, Supervision, Funding acquisition, Writing - original draft, Project administration, Writing - review and editing

### Author ORCIDs
Congyan Zhang http://orcid.org/0000-0002-0737-3876
Randy Schekman http://orcid.org/0000-0001-8615-6409

### Decision letter and Author response
Decision letter https://doi.org/10.7554/eLife.84391.sa1
Author response https://doi.org/10.7554/eLife.84391.sa2

## Additional files

### Supplementary files
• MDAR checklist

### Data availability
All data generated or analysed during this study are included in the manuscript and supporting file. Source data files have been provided for Figure 1B; Figure 1- figure supplement 1A and 1B; Figure 3- figure supplement 2A, 2B, 2C, 2D, 2E, 2F, 2G, 2H, 2I, and 2J; Figure 7- figure supplement 1A, 1B, 1D, 1F, 1G, and 1H; Figure 8- figure supplement 1D; Figure 9B and 9D.

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
