## [Editor Report]

This fundamental work extends and in substantive ways introduces new concepts in the mode of communication between cells. Molecules pass from one cell to another through membrane tubules and the investigators show here convincingly that this occurs exclusively through the physical connection of the open ended tubules and not through exosomes; this process requires syncytin proteins, and the functionality of the protein transferred is retained.

---

## [Decision Letter]

**Decision letter after peer review:**

Thank you for submitting your article "Syncytin-mediated open-ended membrane tubular connections facilitate the intercellular transfer of cargos including Cas9 protein" for consideration by *eLife*. Your article has been reviewed by 3 peer reviewers, one of whom is a member of our Board of Reviewing Editor, and the evaluation has been overseen by Suzanne Pfeffer as the Senior Editor. The following individual involved in review of your submission has agreed to reveal their identity: Gal Haimovich (Reviewer #2).

Essential revisions:

One reviewer is more extensive in the comments which need to be addressed as best as is possible. Most of these are clarifications but the reviewer indicates that more convincing evidence is needed.

1) Clarify text to address questions raised by reviewer 2.

2) An experiment suggested by reviewers 2 and 3 is to use immunofluorescence to localize the presence of the fusion determinants on the membrane. Perhaps this will clarify the differences in receptivity in the cell line. There is some confusion here as to the level of the protein and the corresponding ability to form open connections.

3) A clarification of the differences between the role of these observed membrane tubules and the TNTs reported in other works in transfer of molecules (eg mRNA).

Is it just a thickness difference?

*Reviewer #1 (Recommendations for the authors):*

It would be more generalizable to living systems to look at primary cells rather than cell lines.

*Reviewer #2 (Recommendations for the authors):*

1. Related to public comment 1: If str-CD63 is absent from the transiently transfected cells, then an additional comparison should be to transiently transfect SBP-Cas9-GFP to cells expressing myc-str-CD63 and the Nluc reporter, in the presence or absence of biotin. This will show whether the str-SBP interaction is inhibiting Cas9 nuclear import and if biotin actually releases SBP-Cas9-GFP for nuclear import (as an additional control to the co-IP in Figure 1 – supp figure 1A).

2. Related to public comment 2:

c. Assuming Syncytin-2 is not required for TNTs transfer, a simple experiment to distinguish TNT and membrane tubules will be to track TNT formation, or transfer of known TNTs cargos (e.g. vesicles, mitochondria, mRNA) in WT vs Syncytin K/O.

d. To rule out the possibility that the observed phenotype is due to mRNA transfer through TNTs, the authors could transfect the acceptor cells with siRNA against the SBP-Cas9-GFP mRNA. The phenotype should not change if mRNA transfer does not play a role here.

e. At the very least, the authors should discuss these possibilities and why they think TNTs do not play a role in the Cas9/split-GFP transfer since both the protein and mRNA can transfer through TNTs, yielding the observed results.

3. Related to public comment #3: Split GFP assay

a. A FACS analysis of mixed (not co-cultured) cells similar to the one presented in Figure 2C, F should be shown.

b. Knocking-down Syncytin for the split-GFP assay will strengthen the claim that both assays measure the same process.

4. Related to public comment #4c: The authors should show live imaging of single-culture acceptor cells to see the background in the mCherry and GFP channel.

5. Related to public comment #5: I think a relatively simple experiment to examine the function of Syncytin in HEK293T will be to determine the syncytin localization in MDA vs HEK cells (e.g. at cell surface, contact sites, membrane tubules). I think it can greatly improve the authors' claim on the role of Syncytin and resolve the HEK problem.

6. The authors show that transferred Cas9, as detected by immunofluorescence (Figure 1H), is maintained in vesicular formations in the acceptor cells – probably endosomes or lysosomes as previously demonstrated for EV uptake (e.g. PMID: 27114500), but the authors do not provide more details on these structures. Is the IF signal for CD63 represents the endogenous CD63 in acceptor cells or myc-str-CD63 coming from the donor cells? The figure legend reads "anti-CD63" antibody suggesting endogenous, but no such antibody appears in the list in Materials and methods section, so maybe the authors meant anti-myc? Furthermore, co-IF with antibodies against early/late endosome/lysosome markers could clarify at which stage Cas9 gets "stuck".

7. The authors perform the co-culture experiments on a time scale of days. From my experience with TNT-mediated mRNA transfer, this is most likely because the initial culture was at 10% confluence – with low chance of cell-cell contact. Have the authors tried looking at transfer in a 1:1 or 10:1 culture plated already at 60-70% confluence? This could yield more interactions that could be detected and studied.

8. Latrunculin – my experience suggests it is very toxic to cells (I used 100-200nM) and typically the cells die within several hours. Yet here the authors treated the cells for 3-6 days. Furthermore, such a treatment will likely affect cell motility and proliferation – thus limiting not only formation of tubules but also decrease the probability for cell-contacts. The authors should state if they checked for cell viability and if cell proliferation was affected – thus reducing confluence. Experiments with high-confluence at initiation and therefor reduced co-culture duration (item #2 above) could help with the toxicity by reducing time of exposure to the drug.

9. The authors spend two paragraphs in the discussion to suggest a role in cancer (transformed) cells vs non-transformed cells. However, the only example of non-transformed cells they have are the HEK293T cells – and the lack of transfer can be a unique issue of this particular cell line. I suggest to limit the discussion of this subject or provide data from at least one more non-transformed cell line (which also expresses Syncytin).

10. The section on the split GFP is very confusing and hard to follow – for example I needed to read several times until I understood what "Q2-4/Q4-4+Q2-4". And what are the black-labeled cell populations and how are they different from the green or purple populations in the dot-plots? It will also be helpful to label the axes with the relevant fluorescent protein measured, rather the filter name. I suggest using simpler labeling and more explanations for better clarity.

11. The authors suggest in their model that Syncytin and MFSD2A on opposite cells "meet" at the tip of membrane tubules. I believe imaging of Syncytin and MFSD2A proteins will help support this model and may resolve the inconsistent result with HEK293T cells (see public comment #5).

*Reviewer #3 (Recommendations for the authors):*

1. A formin inhibitor could be used to test their potential involvement.

2. Statistical analyses may be used to assess the occurrence of cargo transfer along a broad cell-cell contact zone.

3. Antibody staining of syncytins and/or MFSD2A could be performed (if antibodies are available) to detect the presence of these proteins on the cell surface in difference conditions.

---

## [Author Response]

Essential revisions:1) Clarify text to address questions raised by reviewer 2.

Response to each question as below.

2) An experiment suggested by reviewers 2 and 3 is to use immunofluorescence to localize the presence of the fusion determinants on the membrane. Perhaps this will clarify the differences in receptivity in the cell line. There is some confusion here as to the level of the protein and the corresponding ability to form open connections.

Since the commercial antibodies for syncytins are not good for immunofluorescence, we relied on cells transfected with fluorescent protein-tagged syncytins. The results show that syncytins partially localized to the cell surface (Figure 8—figure supplement 1A-C), including at the junction between a tubule and the target cell surface (Figure 8—figure supplement 1E, F). We found that overexpressed syncytin-1 and -2 induced cell fusion in HEK293T cells, consistent with the cell surface localization of at least a fraction of these proteins. Nonetheless, the basal level of syncytins at the surface of HEK293 cells may be inadequate to promote open-ended tubular connections in cultures of this cell line. Answering this question will require more sensitive probes to detect the endogenous syncytins.

As an independent test of cell surface localization, we used surface biotinylation to show that a fraction of the syncytins can be labeled externally (Figure 8—figure supplement 1D). This fraction shows evidence of proteolytic processing consistent with furin cleavage whereas the overwhelming majority of transfected syncytins detected in a blot of lysates suggests that most remains in the unprocessed precursor form, consistent with the punctate and reticular fluorescence images (Figure 8—figure supplement 1A-C).

3) A clarification of the differences between the role of these observed membrane tubules and the TNTs reported in other works in transfer of molecules (eg mRNA).Is it just a thickness difference?

The results in this study cannot distinguish between roles for TNTs or thick membrane tubules. We are not yet aware of distinct requirements for the formation of these two tubular connections. It may be that TNTs contain actin only, whereas thick tubules contain both actin and microtubules (ref: PMID: 17142745), which may result in the transfer of different cargos, for example, the transfer of microtubule-mediated cargos through thick tubular connections. Such functional distinctions bear further exploration thus we have been careful avoid implying a role for one or the other, except insofar as our visual detection has focused on the more obvious role of thick tubules in intercellular transfer.

Reviewer #2 (Recommendations for the authors):1. Related to public comment 1: If str-CD63 is absent from the transiently transfected cells, then an additional comparison should be to transiently transfect SBP-Cas9-GFP to cells expressing myc-str-CD63 and the Nluc reporter, in the presence or absence of biotin. This will show whether the str-SBP interaction is inhibiting Cas9 nuclear import and if biotin actually releases SBP-Cas9-GFP for nuclear import (as an additional control to the co-IP in Figure 1 – supp figure 1A).

We added new data in Figure 1—figure supplement 1D which suggests that CD63-tethering itself does not affect Cas9 function.

2. Related to public comment 2:c. Assuming Syncytin-2 is not required for TNTs transfer, a simple experiment to distinguish TNT and membrane tubules will be to track TNT formation, or transfer of known TNTs cargos (e.g. vesicles, mitochondria, mRNA) in WT vs Syncytin K/O.d. To rule out the possibility that the observed phenotype is due to mRNA transfer through TNTs, the authors could transfect the acceptor cells with siRNA against the SBP-Cas9-GFP mRNA. The phenotype should not change if mRNA transfer does not play a role here.e. At the very least, the authors should discuss these possibilities and why they think TNTs do not play a role in the Cas9/split-GFP transfer since both the protein and mRNA can transfer through TNTs, yielding the observed results.

The results in this study do not rule out a role for TNTs in intercellular transfer nor do we suggest that syncytin-2 is not required for TNT-mediated transfer. This point is clarified in the discussion.

3. Related to public comment #3: Split GFP assaya. A FACS analysis of mixed (not co-cultured) cells similar to the one presented in Figure 2C, F should be shown.b. Knocking-down Syncytin for the split-GFP assay will strengthen the claim that both assays measure the same process.

A FACS analysis of mixed (not co-cultured) cells was included. The results in this study do not rule out a role for TNTs in the transfer of both Cas9 and split-GFP.

4. Related to public comment #4c: The authors should show live imaging of single-culture acceptor cells to see the background in the mCherry and GFP channel.

Images of single-culture acceptor cells have been added in Figure 4—figure supplement 1.

5. Related to public comment #5: I think a relatively simple experiment to examine the function of Syncytin in HEK293T will be to determine the syncytin localization in MDA vs HEK cells (e.g. at cell surface, contact sites, membrane tubules). I think it can greatly improve the authors' claim on the role of Syncytin and resolve the HEK problem.

We have performed new visualization and cell surface labeling experiments to show that syncytins appear to be partially localized at the cell surface of MDA-MB-231 cells, and even at sites of cell-cell contact (Figure 8—figure supplement 1). We also find that overexpressed syncytin-2 localizes to the plasma membrane of HEK293T cells to induce cell fusion. As stated above, the problem remains to explain why basal levels of expression of syncytins are inadequate to promote the formation of open-ended tubular connections.

6. The authors show that transferred Cas9, as detected by immunofluorescence (Figure 1H), is maintained in vesicular formations in the acceptor cells – probably endosomes or lysosomes as previously demonstrated for EV uptake (e.g. PMID: 27114500), but the authors do not provide more details on these structures. Is the IF signal for CD63 represents the endogenous CD63 in acceptor cells or myc-str-CD63 coming from the donor cells? The figure legend reads "anti-CD63" antibody suggesting endogenous, but no such antibody appears in the list in Materials and methods section, so maybe the authors meant anti-myc? Furthermore, co-IF with antibodies against early/late endosome/lysosome markers could clarify at which stage Cas9 gets "stuck".

Here we used anti-CD63 and have added that in the list in Materials and methods section. We observed endogenous CD63 in the PBS control images, suggesting that Cas9 is at least partially localized in endosome related vesicles.

7. The authors perform the co-culture experiments on a time scale of days. From my experience with TNT-mediated mRNA transfer, this is most likely because the initial culture was at 10% confluence – with low chance of cell-cell contact. Have the authors tried looking at transfer in a 1:1 or 10:1 culture plated already at 60-70% confluence? This could yield more interactions that could be detected and studied.

We tried 10:1 culture plated at 60-70% confluence, and cultured for short times but found less transfer (1d vs 3d in Figure 2—figure supplement 1A).

8. Latrunculin – my experience suggests it is very toxic to cells (I used 100-200nM) and typically the cells die within several hours. Yet here the authors treated the cells for 3-6 days. Furthermore, such a treatment will likely affect cell motility and proliferation – thus limiting not only formation of tubules but also decrease the probability for cell-contacts. The authors should state if they checked for cell viability and if cell proliferation was affected – thus reducing confluence. Experiments with high-confluence at initiation and therefor reduced co-culture duration (item #2 above) could help with the toxicity by reducing time of exposure to the drug.

For latrunculin treatment, we used latrunculin A at 40, 80, 200 nM and latrunculin B at 1, 2.5, 5 µM. We checked the cell viability, and found that high concentrations of latrunculin (such as latrunculin A at 200 nM, and latrunculin B at 5 µM) affected cell growth. Thus for the high concentrations, at initiation we used a higher cell density (a little more than DMSO) and cultured for longer times until near confluence to diminish the toxic effect. The Nluc/Fluc signal detected after co-culture suggested that more than half of the cells remained viable during drug treatment.

9. The authors spend two paragraphs in the discussion to suggest a role in cancer (transformed) cells vs non-transformed cells. However, the only example of non-transformed cells they have are the HEK293T cells – and the lack of transfer can be a unique issue of this particular cell line. I suggest to limit the discussion of this subject or provide data from at least one more non-transformed cell line (which also expresses Syncytin).

We reduced this part of the Discussion to one paragraph.

10. The section on the split GFP is very confusing and hard to follow – for example I needed to read several times until I understood what "Q2-4/Q4-4+Q2-4". And what are the black-labeled cell populations and how are they different from the green or purple populations in the dot-plots? It will also be helpful to label the axes with the relevant fluorescent protein measured, rather the filter name. I suggest using simpler labeling and more explanations for better clarity.

This is a valuable suggestion to help for clarity. We now label the axes with the relevant fluorescent protein measured. The black-labeled cell populations should be the doublet cells, the plot in the Figure 2C and F were displayed with an all-cell mode.

11. The authors suggest in their model that Syncytin and MFSD2A on opposite cells "meet" at the tip of membrane tubules. I believe imaging of Syncytin and MFSD2A proteins will help support this model and may resolve the inconsistent result with HEK293T cells (see public comment #5).

Imaging data for syncytin and MFSD2A proteins are now included. We find that syncytins partially localize at cell surface of MDA-MB-231 cells, including at cell-cell contact sites (Figure 8—figure supplement 1). MFSD2A partially is localized to the plasma membrane of HEK293T cells.